# Identifying source regions of air masses sampled at the tropical high-altitude site of Chacaltaya using WRF-FLEXPART and cluster analysis

Diego Aliaga[1], Victoria A. Sinclair[1], Marcos Andrade[2,3], Paulo Artaxo[4], Samara Carbone[5], Evgeny Kadantsev[1,6], Paolo Laj[1,7], Alfred Wiedensohler[8], Radovan Krejci[9], and Federico Bianchi[1]

[1]Institute for Atmospheric and Earth System Research /Physics, Faculty of Science, University of Helsinki, Helsinki, 00014, Finland
[2]Laboratory for Atmospheric Physics, Institute for Physics Research, Universidad Mayor de San Andrés, La Paz, Bolivia
[3]Department of Atmospheric and Oceanic Sciences, University of Maryland, College Park, MD, USA
[4]Institute of Physics, University of Sao Paulo, São Paulo, Brazil
[5]Federal University of Uberlândia, Agrarian Sciences Institute, Uberlândia, MG, Brazil
[6]Finnish Meteorological Institute, Helsinki, 00101, Finland
[7]Université Grenoble Alpes, CNRS, IRD, Grenoble INP, Institut des Géosciences de l'Environnement, Grenoble, France
[8]Leibniz Institute for Tropospheric Research, Permoserstr. 15, 04318 Leipzig, Germany
[9]Department of Environmental Science & Bolin Centre of Climate Research, Stockholm University, Stockholm 10691,Sweden

Correspondence: Victoria Sinclair (victoria.sinclair@helsinki.fi)

**Abstract.** Observations of aerosol and trace gases in the remote troposphere are vital to quantify background concentrations and identify long term trends in atmospheric composition on large spatial scales. Measurements made at high altitude are often used to study free tropospheric air however such high-altitude sites can be influenced by boundary layer air masses. Thus, accurate information on air mass origin and transport pathways to high altitude sites is required. Here we present a new method, based on the source-receptor relationship (SRR) obtained from backwards WRF-FLEXPART simulations and a k-means clustering approach, to identify source regions of air masses arriving at measurement sites. Our method is tailored to areas of complex terrain and to stations influenced by both local and long-range sources. We have applied this method to the Chacaltaya (CHC) GAW station (5240 m a.s.l., 16.35° S, 68.13° W) for the 6-month duration of the "Southern hemisphere high altitude experiment on particle nucleation and growth" (SALTENA) to identify where sampled air masses originate and to quantify the influence of the surface and the free troposphere. A key aspect of our method is that it is probabilistic and for each observation time, more than one air mass (cluster) can influence the station and the percentage influence of each air mass can be quantified. This is in contrast to binary methods, which label each observation time as influenced either by boundary layer or free troposphere air masses. Air sampled at CHC is a mix of different provenance. We find that on average 9 % of the air, at any given observation time, has been in contact with the surface within 4 days prior to arriving at CHC. Furthermore, 24 % of the air has been located within the first 1.5 km above ground level (surface included). Consequently, 76 % of the air sampled at CHC originates from the free troposphere. However, pure free-tropospheric influences are rare and often samples are concurrently influenced by both boundary-layer and free-tropospheric air masses. A clear diurnal cycle is present with very

few air masses that have been in contact with the surface being detected at night. The 6-month analysis also shows that the most dominant air mass (cluster) originates in the Amazon and is responsible for 29 % of the sampled air. Furthermore, short-range clusters (origins within 100 km of CHC) have high temporal frequency modulated by local meteorology driven by the diurnal cycle whereas the mid- and long-range clusters' (>200 km) variability occurs on timescales governed by synoptic-scale dynamics. To verify the reliability of our method, in-situ sulfate observations from CHC are combined with the SRR clusters to correctly identify the (pre-known) source of the sulfate: the Sabancaya volcano located 400 km northwest from the station.

# 1   Introduction

Traditionally, high-altitude measurement sites are used to study the remote atmosphere and the interactions between the free troposphere (FT) and the planetary boundary layer (PBL). These sites provide the opportunity to have long-term in situ observations of the free troposphere with high time resolution (Collaud Coen et al., 2018) as opposed to the short duration and inherent transient nature of airborne measurement campaigns. Observations of aerosol and trace gases in the FT are of great scientific value in terms of understanding long-range transport and atmospheric chemistry, quantifying background concentrations, and observing long term changes in the composition of the atmosphere (Laj et al., 2009). Nevertheless, it is well known that high-altitude mountain sites can be influenced by boundary-layer air.

The planetary boundary layer is the lowest part of the atmosphere and is in direct contact with the Earth's surface. The majority of natural, and especially anthropogenic, aerosols and pollutants are emitted at the surface and thus directly released into the PBL. The concentrations of gases and aerosols and their residence times depends on the structure of the PBL. Over land the PBL has a pronounced diurnal cycle with deep, well-mixed boundary layers typically observed during the day and shallow, stable boundary layers occurring at night. In complex terrain, thermally driven winds develop (e.g. slope and valley winds, De Wekker and Kossmann, 2015) during the day and can transport aerosols and pollutants from valley bottoms to high-altitude sites. Additionally, complex mountain meteorological processes such as orographic lifting can also transport PBL air to high altitude.

Therefore, high-altitude sites can be influenced by local boundary-layer air and free tropospheric air where the latter may have undergone long-range transport due to stronger upper-level winds. This means that observations must be carefully screened and analysed in synergy with many parameters to understand the dynamics and diurnal cycle of individual sites. Consequently, understanding the history of air masses sampled at mountain-top sites and related chemical composition is not an easy exercise. Since the chemical and physical composition of the sampled air masses is, in general, inherently related to its path through the atmosphere (Fleming et al., 2012), having good methods to describe the history of the sampled air masses increases the value of measurements. Under these circumstances, the observations can then be treated as samples from different parts of the atmosphere both in the vertical (PBL vs FT) and the horizontal (short-range vs long-range contributions) domain.

Much of the analysis of ground-based atmospheric composition observations is accompanied by studies of what is known as "climatological pathways" with the aim of mapping the probability of certain air masses reaching the station and identifying the sources and processes influencing specific types of air masses (Fleming et al., 2012). There are various ways of identifying air mass source regions. A popular and easy-to-apply approach to identify whether the measured air mass originates from the PBL or FT is to consider the presence of specific tracers such as Radon-222 (Griffiths et al., 2014) or the ratio of carbon monoxide to oxidized nitrogen species (CO/NO$_y$, Herrmann et al., 2015). This tracer based approach has the limitation that it cannot easily resolve horizontal source areas and only resolves two layers in the vertical. Alternative methods to identify source regions, which utilizing meteorological observations or numerical models, range from simply looking at the local wind direction to single trajectory analysis, Lagrangian dispersion models and even chemical transport models. The choice of method to link the atmospheric composition to the air mass history largely depends on where the expected sources are.

If the sources are predominately local, using local winds observations as a proxy for air mass history may suffice and this approach has been taken by e.g. de Foy et al. (2008) and Salisbury et al. (2002). The major weakness of using in-situ wind measurements as an indicator of air mass history is that we cannot assume that the local measurements are representative of a larger region or of the synoptic-scale flow. This is particularly important for regions with complex topography in which the wind speed and direction at the surface may differ drastically to the wind aloft. Alternatively, if sources are remote, a low-resolution modelling approach may be more appropriate to identify air mass history. Such approaches have been applied by Brattich et al. (2020) at Mt. Cimone in Italy and by Sturm et al. (2013) at Jungfraujoch in Switzerland.

Single back trajectory models, often driven by low-resolution meteorological input data, have been widely used to gain insight into the sampled air mass history at high-altitude sites (Keresztesi et al., 2020 [Eastern Carpathians, Romania]; Brattich et al., 2020 [Mt. Cimone, Italy]; Ghasemifard et al., 2019 [Schneefernerhaus, Germany]; Bolaño-Ortiz et al., 2019 [Central Andes, Chile]; Qie et al., 2018 [Mount Tai, China]; Ou-Yang et al., 2017 [Mt. Fuji, Japan]; Chauvigné et al., 2016 [Mt. Chacaltaya, Bolivia]; Brattich et al., 2015 [Mt. Cimone, Italy]; Gratz et al., 2015 [Mt. Bachelor, U.S.A.]; Ou-Yang et al., 2014 [Mt. Lulin, Taiwan]; Putero et al., 2014 [Askole, Pakistan]; Tositti et al., 2013 [Mt. Cimone, Italy]; Cheng et al., 2013 [Mt. Lulin, Taiwan]; and Fleming et al., 2012 references). Advantages of this method are that it is computationally efficient, easy to perform, and for individual case studies the output is simple to interpret. These studies primarily consider the horizontal, large-scale flow obtained by the single trajectory model. This becomes problematic at high-altitude sites where the influence of the complex topography is known to have a clear effect on the interaction with the convective boundary layer (Serafin et al., 2018). Another disadvantage of single back trajectories, as shown by Stohl et al. (2002), is that they do not account for the filamentation and backward volume growth of the finite sampled air masses.

Lagrangian dispersion models have also been used to identify source areas at high-altitude sites (Ubl et al., 2017 [Zeppelin-fjellet, Norway]; Cécé et al., 2016 [Guadeloupe archipelago, Caribbean]; Lopez et al., 2015 [Puy de Dome, France]; Sturm et al., 2013 [Jungfraujoch, Switzerland]; Brunner et al., 2012 [Jungfraujoch, Switzerland]; Conen et al., 2012 [Jungfraujoch, Switzerland]; Hirdman et al., 2010 [Alert, Barrow and Zeppelin; Arctic]; and de Foy et al., 2009 [Mexico City Metropolitan Area, Mexico]). trajectory models by accounting for the deformation and stochastic dispersion of air masses due to turbulent mixing and convection (Stohl et al., 2002). A notable downside of dispersion models, particularly when applied to studies covering many months, is that they produces a large amount of output which is quite complicated to understand and interpret. Therefore, additional post-processing and automated, objective analysis methods need to be applied to this output to extract the maximum amount of information while ensuring the resulting dataset is user-friendly. One way of post-processing dispersion model output is to perform a cluster analysis, where statistical methods can be used to differentiate different source regions.

Cluster analysis is a multivariate technique used to classify elements into groups in a way that maximizes the similarity (by a predefined metric) within members of a group while also maximizing the dissimilarity across groups. Clustering has been extensively used in studies that aim to classify air mass history. In the case of single trajectory studies, the goal is to group trajectories into ensembles that follow a similar pathway (Kassomenos et al., 2010 and references therein). In dispersion models, clustering analysis has been applied both to classify the retroplume of the particles (e.g. Stohl et al., 2002) and also to classify the regional footprint of the particles (e.g. Sturm et al., 2013 and Paris et al., 2010). However, these studies mostly

assume that high-altitude sites are also background sites, and therefore are mostly interested in the contribution of long-range transport sources. This means that the meteorological data used to drive the dispersion models may have low spatial resolution since the local sources are assumed to be negligible. However, there is still a lack of classification in the vertical dimension and accountability for the influence of short- and long-range transport simultaneously which is of special relevance for locations where short-range sources are equally relevant to more distant sources.

One such high-altitude site is the GAW (Global Atmospheric Watch) Chacaltaya (CHC) atmospheric research station (5240 m a.s.l., 16.35° S, 68.13° W) located 20 km from the metropolitan area of La Paz / El Alto but ~1.6 km higher in altitude than the centre of La Paz. For a detailed description of the site see Chauvigné et al. (2019) and Wiedensohler et al. (2018). Measurements of reactive and greenhouse gases as well as aerosol optical, chemical and physical properties are routinely monitored at the station following the GAW recommended procedures (Laj et al., 2020). At this station, in the context of the SALTENA (Southern hemisphere high altitude experiment on particle nucleation and growth) campaign (Bianchi et al., 2021), state-of-the-art instruments that measure aerosol chemical and physical properties were deployed to complement on-going long-term observations. The intensive measurements took place between December 2017 and June 2018 (covering both wet and dry seasons). The unique location of the station in the under-sampled southern hemisphere enabled us to study a mixture of pristine air masses from the Amazon Basin loaded with biogenic emissions, regional background air masses from the Altiplano perturbed by volcanic activity, and, marine air masses from the Pacific Ocean. In addition, strong anthropogenic influence from the La Paz / El Alto metropolitan area was sampled. This wide range of potential source areas, along with complex mountain meteorology, and highly detailed observations of the physical and chemical properties of aerosol and trace gases, means that a comprehensive meteorological analysis, beyond what is the typical performed for aerosol measurement campaigns, is required.

The overall objective of this study is to develop, and apply to CHC, a new method to identify air mass source regions which is valid for high-altitude stations that are influenced both by local and long-range sources and where the vertical classification of sources is as relevant as horizontal segregation. An outline and overview of the method is given in Fig. 1. The first aim of this study is to use a regional meteorological model (Weather Research and Forecasting model—WRF) in combination with a Lagrangian dispersion model (FLEXible PARTicle dispersion model—FLEXPART) to create a high-resolution data set of source areas for CHC at hourly resolution (steps 1 and 2 in Fig. 1). The second aim is to develop a new method, based on cluster analysis, to transform the complex, very large and multi-dimensional dataset into a user-friendly dataset of air mass source regions (steps 3 to 7). The third aim is to document the characteristics of the identified source areas (clusters) which will enable the dataset produced here to be applied in forthcoming studies on the chemical composition measurements made during the unique high-altitude SALTENA campaign. The fourth and final aim is to demonstrate the strength and simplicity of the classification results from our method which we do by using them to confirm a well-known source of sulfate emissions that were measured at the CHC station.

The remainder of this study is structured as follows. In section 2, the meteorological model and the Lagrangian dispersion model are described (Fig. 1, steps 1 and 2). The newly developed clustering method is described in section 3 (steps 3 to 7 in Fig. 1). Additional diagnostics are presented in section 4. The spatial distribution of dispersion model output are presented in

**Figure 1.** Flowchart describing the method's steps. The steps are divided into three groups: modelling, pre-processing and clustering. SRR refers to the source-receptor relationship (explained more in section 3).

section 5.1. The relative contribution of the surface, the PBL and the FT to CHC are described in section 5.2. The characteristics of the identified source regions are discussed in  sections 5.3 and 5.4. An example indicating that the method works well is shown in section 5.5. A discussion on the results and recommendations are presented in section 6. Finally, the conclusions are presented in section 7.

## 2   High resolution meteorological modelling and backward dispersion simulations

### 2.1   High resolution meteorological modelling (step 1)

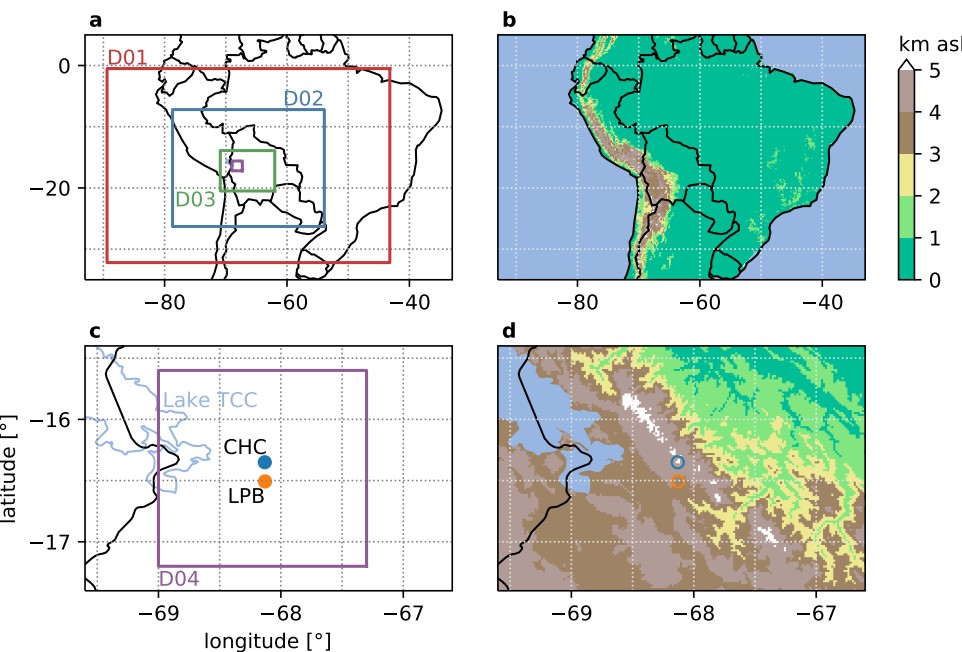

**Figure 2.** Overview of the studied region and the WRF model domains. Panel a) shows the location of the 4 nested domains (D01–DO4), b) shows the topography in kilometers above sea level (km a.s.l.) of the whole domain, c) is a zoomed in version of a) and also shows the location of the Chacaltaya station (CHC, 5.2 km a.s.l., blue dot), La Paz / El Alto metropolitan area (LPB, 3.6 km a.s.l., orange dot) and Lake Titicaca (TCC, 3.9 km a.s.l., blue outline). Panel d) shows the topography (km a.s.l.) in the area closest to CHC.

To generate a high-resolution, gridded dataset of meteorological variables that can be used to drive a Lagrangian dispersion model, we used the Weather Research and Forecasting (WRF) model version 4.0.3. WRF is a state-of-the-art, non-hydrostatic, regional numerical weather prediction model that is used operationally and for research. Here we perform one 6-month long continuous simulation starting on 2017-12-06 and ending on 2018-05-31. As the simulation and subsequent analysis is only 6 months long, this study provides detailed air-mass information for the duration of the intensive period rather than a climatological description which would require a multiple-year study. Four nested domains are used (D01–D04) and their locations

are shown in Fig. 2a and c. The outermost and largest domain (D01) has a grid spacing of 38 km whereas the innermost and smallest domain (D04) has a grid spacing of 1 km. Full details of the domains are given in Table S1. One-way nesting is used: the outer domain provides boundary data for the inner nest but the inner nest does not provide any feedback to the outer domain. To ensure the long simulation remains close to reality throughout the 6-month period, the outer domain is nudged (i.e. analysis nudging) to the boundary conditions every 6 hours.

The initial and boundary conditions were taken from the NCEP Climate Forecast System Version 2 (Saha et al., 2011, 2014) with a temporal resolution of 6 hours, a 0.5° horizontal resolution and 64 sigma-pressure hybrid layers. The model topography was obtained from the Global multi-resolution terrain elevation data model (Danielson and Gesch, 2011) with a resolution of ∼1 km. We use the following parameterizations: microphysics is parameterized by the Goddard Scheme; cumulus convection is parameterized by the Grell–Freitas Ensemble Scheme in D01 and D02 and no cumulus parameterization is used for D03 and D04; the short and long-wave radiation is parameterized by the New Goddard Shortwave and Longwave Schemes; the Planetary Boundary Layer (PBL) Physics are represented by the Mellor–Yamada–Janjić Scheme (MYJ); and the land surface model is the Unified Noah Land Surface Model.

The surface temperature of Lake Titicaca was manually prescribed to monthly means obtained from Pillco Zolá et al. (2019) since values prescribed by WRF were unrealistically low. This is most likely due to the height of the lake (around 3.8 km a.s.l.) and the assumptions made by WRF when interpolating surface temperature of lakes from adjacent sea surface temperature (a similar lake-temperature issue is reported by Valerio et al., 2017).

There are some limitations to the WRF simulation, primarily related to the complexity of the terrain. The 1-km resolution inevitably smooths the topography in comparison to reality which can affect the slope angles and furthermore affect the simulated thermally-driven winds. However, temperature, and precipitation comparisons with in situ observations at CHC show reasonable agreement. Figure S1 shows the 6-month timeseries of the modelled and observed hourly temperature, daily mean temperature and daily accumulated precipitation. Using this data, basic error metrics were computed. For the hourly temperature data, the mean bias, mean absolute error and root mean square error are -0.42° C, 1.35° C and 1.73° C respectively. For the daily accumulated precipitation, we compiled contingency tables with different precipitation thresholds and computed the accuracy (see the supplementary material for details of these calculations). For a threshold of 1 mm, the accuracy (i.e. fraction of correct forecasts) is 0.65. Additional evaluation using 0 mm and 5 mm thresholds, along with the number of hits, misses, false alarms and correct negatives are shown in Table S2. Furthermore, an evaluation of the WRF-simulated monthly accumulated precipitation for December and May at a number of stations near CHC is presented by Bianchi et al. (2021).

## 2.2 Backward dispersion simulations (step 2)

The FLEXible PARTicle dispersion model (FLEXPART) is a Lagrangian transport and dispersion model which can be used for both forward and backward simulations. We used version FLEXPART-WRF_v3.3.2 (Brioude et al., 2013) to perform backward simulations and thus to determine the source regions of air masses arriving at CHC. The FLEXPART simulations were driven using the meteorological output from the 6-month WRF simulation. Output from all four of the WRF output domains was used

and this was available at a temporal resolution of 15 minutes. This high temporal resolution is a clear advantage over using reanalysis data which at best is only available once per hour.

In the FLEXPART simulations, we continuously release 20 000 particles per hour from CHC from 2017-12-06 until 2018-05-31 and compute their back trajectories for 4 days. The particles, passive air tracers, are released in a 10-m deep layer which extends from 0 to 10 meters a.g.l. and over a 2 x 2 km square centred around CHC. With the choice of 4 days as the (backwards) simulation time, the average median particle spends 94% of its residence time within the domain D01.

When FLEXPART is run in backward mode, it calculates the emission sensitivity response function, also referred to as the source-receptor relationship (SRR), on a user-specified three-dimensional longitude-latitude-height grid. The output of FLEXPART can be in different units and here we configure the model so that the source (IND_SOURCE=2) and the receptor (IND_RECEPTOR=2) are in mass mixing ratio mode and therefore the output (SRR) is in units of seconds (see Table 1 in Eckhardt et al., 2017). We choose mass mixing ratios so that the SRR matrix is not affected by pressure variations in the 3D domain.

FLEXPART also permits two user-specified nested output domains with the inner domain closest to the release site (i.e. the receptor), having a higher resolution than the outer domain. We make use of this functionality and specify the first FLEXPART output domain to have the same geographic extent as D03 in the WRF simulation but the spatial resolution (1 km) as D04 in the WRF simulation. Our second, outer FLEXPART output domain covers the same region as D01 in the WRF simulation and has the same resolution (9.5 km) as D02 in WRF. In the vertical direction, we specify the FLEXPART output grid to have 30 uniform levels each 500 m deep extending from the surface to 15 km a.g.l. The rationale for using uniform level-spacing is explained in the Appendix, section A1.

FLEXPART also contains options for how turbulence and convection are included in the simulations. We take the values of the PBL height, surface sensible heat flux, and the friction velocity directly from the WRF simulation (SCF_OPTION=1). Turbulence is parameterized using the Hanna scheme (Hanna, 1984) as used in FLEXPART-ECMWF/GFS ( TURB_OPTION=1) and we assume skewed rather than Gaussian turbulence in the convective boundary layer (CBL=1). Deep convection is also parameterized (LCONVECTION=1).

## 3 Pre-processing and clustering of the FLEXPART output

Here we describe the core of our new method, namely the log-polar grid transformation (Fig. 1, step 3), the grid cell pre-processing (step 4), the iterative k-means clustering algorithm (step 5), the silhouette scoring (step 6) and finally the selecting the optimal number of clusters (step 7). Additional complementary technical details are presented in Appendix A.

### 3.1 Log-polar grid transformation (step 3)

The output from FLEXPART is the source-receptor relationship (SRR) which is related to the particles' residence time in the output 3D grid cells. These cells are defined by a regular longitude ($x$), latitude ($y$), and height ($z$) grid. In addition to the 3 spatial dimensions, the SRR has 2 time dimensions; the release-time ($t$, the time when the particles arrive at the release

location); and the backwards-time ($\tau$, the amount of time before the release time which varies from 0 to 96 hours in our case). Thus, the SRR can be written as the 5-dimensional matrix $\mathbf{SRR}_{xyzt\tau}$. The SRR is processed to remove the $\tau$ time dimension. This is achieved by summing over $\tau$:

$$\mathbf{SRR}_{xyzt} = \sum_{\tau=0}^{96h} \mathbf{SRR}_{xyzt\tau} \tag{1}$$

where the outcome of this step, $\mathbf{SRR}_{xyzt}$, is a 4-dimensional array. The $\tau$ dimension is removed as primarily we want to
determined where the particles spent time in 4 days prior to arriving at CHC, and not when (during the past 4 days) they spent it in each location.

Furthermore, the output SRR is on 2 nested grids (Fig. 3a), which means we obtain high-resolution information for regions that are near the receptor and lower resolution information for more distant locations. The rationale is that, on one hand, near the receptor, the higher resolution provides better detail on potential high-influence sources, for example, in our case, the 20
215    km-away nearby metropolitan area of La Paz / El Alto. On the other hand, far away, a low-resolution grid cell suffices since localized potential source influences are diffused. However, specifying two output grids with different resolution introduces challenges such as the step change in resolution in $\mathbf{SRR}_{xyzt}$ (Fig. 3a): as the SRR is related to the residence time and the number of each particles in a grid cell, smaller cells have fewer particles than larger cells, and typically particles move faster through smaller grid cells than larger grid cells resulting in a smaller residence time and SRR and thus a sharp boundary in the
SRR field exists.

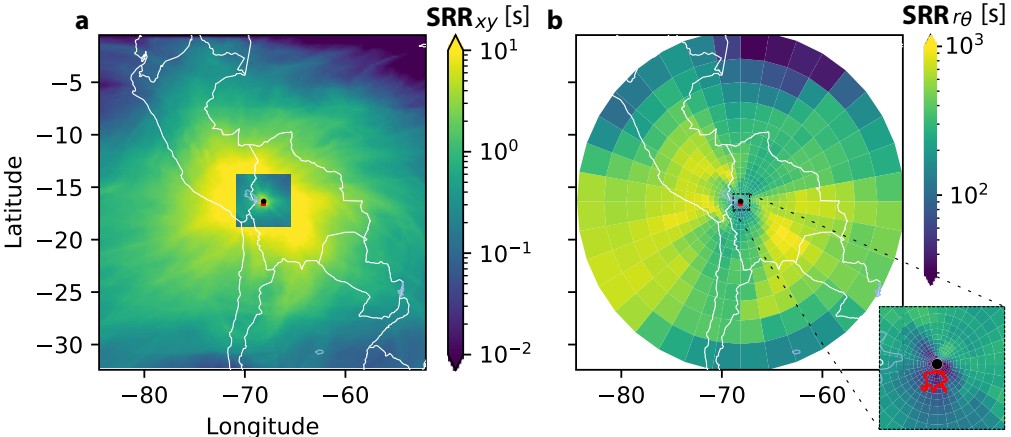

**Figure 3.** Rectangular and log-polar surface plots of the SRR from December 2017 to May 2018. In a) the mean rectangular SRR: $\mathbf{SRR}_{xy} = \sum_z \frac{\sum_t \mathbf{SRR}_{xyzt}}{n_t}$; and in b) the man log-polar SRR: $\mathbf{SRR}_{r\theta} = \sum_z \frac{\sum_t \mathbf{SRR}_{\theta rzt}}{n_t}$. The black dot is CHC and the red polygon is the metropolitan area of La Paz / El Alto.

To overcome these limitations, and more importantly to dramatically reduce the number of grid cells and thus the data volume, we propose the use of a log-polar transform (Sarvaiya et al., 2012) of the coordinate system (Fig. 1, step 3). The new log-polar grid has grid cells which gradually increase in size as the distance from the receptor increases with no sudden

step changes to the resolution. This is also why in Fig. 3b, the SRR does not decrease significantly as the distance from the station is increased (as it does in Fig. 3c). Furthermore, this also implies that the underlying transport patterns are more readily distinguished using the SRR on the log-polar grid than on the equirectangular grid. The regridding process is complex as we had to devise a method that was both computationally efficient and accurate. A detailed explanation of this regridding procedure is given in section A2 of the Appendix and visually illustrated in Figure A1. The results of the regridding process are shown in Fig. 3.b which shows a smooth evolution of the SRR with no boundaries evident at the intersection between the two original nested grids. As a result of the grid transformation, we are left with 33480 log-polar grid cells defined by 36 $\theta$ wedges, 31 $r$ cylinders and 30 $z$ levels: $\mathbf{SRR}_{\theta rzt}$. This is a factor of 328 times smaller than the original number of grid cells. In terms of data volume this, this dramatically reduces the data volume from 186.6 Gb ($\mathbf{SRR}_{xyzt}$) to 0.6 Gb ($\mathbf{SRR}_{\theta rzt}$, see Table S3).

### 3.2 Grid cell pre-processing (step 4)

In general, clustering algorithms benefit from pre-processing of the input dataset which here is the $\mathbf{SRR}_{\theta rzt}$ matrix. This pre-processing modifies the dataset for the sake of grouping the grid cells into clusters—that is assigning each grid cell to a group. In our case, we apply three pre-processing procedures (Fig. 1, step 4): Gaussian filter smoothing, quantile transform and filtering out non-relevant elements. In this pre-processing the 4-dimensional $\mathbf{SRR}_{\theta rzt}$ matrix is also stacked into a 2-dimensional $\mathbf{SRR}'_{\rho t}$ matrix. Specific details about, and additional justification for using, the Gaussian filter smoothing, quantile transform and filtering are presented in the Appendix, section A3. As a side note, it is important to highlight that the processed SRR values ($\mathbf{SRR}'_{\rho t}$) are only used to group the grid cells and not to determine the statistics of each resulting cluster. Instead we combined the cluster numbers with original, non-processed data ($\mathbf{SRR}_{\theta rzt}$) to obtain the results that we present in section 5.

### 3.3 The k-means clustering (steps 5, 6 and 7)

The next step (Fig. 1, step 5) consists of clustering the $\mathbf{SRR}'_{\rho t}$, i.e., the goal is to divide the $\mathbf{SRR}'_{\rho t}$ matrix into $k$ number of groups (clusters), whose individual grid-cell SRR values have a similar evolution in the $t$-dimension. To be clear, the overall objective is to cluster 3D grid cells in the domain based on the SRR contribution that they have over time ($t$) as opposed to clustering the spatial patterns (snapshots) which is the approach taken by Sturm et al. (2013).

In order to perform a cluster analysis on this dataset, we need to define the elements that will be clustered/grouped, the number of clusters, and the features of each element used to determine the group identification. The 33480 log-polar grid cells are taken to be the elements to be clustered. The element's features are the SRR intensities at each release time (although called the release time, it is indeed the arrival time at the destination i.e. the receptor). There are 4248 features for each element, one for each of the hourly releases from 2017-12-06 to 2018-05-31.

We use the k-means (Lloyd, 1982) clustering algorithm due to its generalized use, speed and adequate performance with a large number of elements and medium size number of clusters. Additional details on how the k-means algorithm is applied are presented in section A4 of the Appendix. The result of the k-means clustering is that each grid cell in the 3D log-polar grid is allocated a cluster number.

However, the k-means algorithm does not automatically select the number of clusters $k$ and there is not a right answer for the number of clusters. Too few clusters (e.g. 2) means that no meaningful information is obtained whereas too many clusters (e.g. 100) is impractical and risks overfitting. The optimal number of clusters is usually determined by trying a variety of options and calculating quantitative measures of how similar an element is to its own cluster compared to other clusters. Here we try

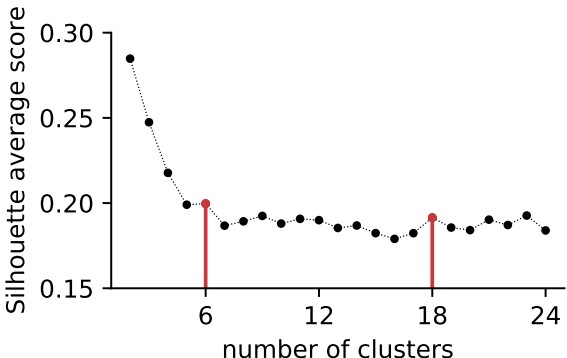

**Figure 4.** Average Silhouette score for the iterative k-means clustering algorithm as a function of number of clusters (from K=2 to 24). Red vertical bars and points show the selected number (6 and 18) of clusters.

2 to 24 clusters and use the silhouette score (Rousseeuw, 1987), which ranges from -1 to +1, to determine the optimal number (Fig. 1 step 6). Large positive values of the silhouette score indicate that the element is well matched within its clusters and poorly matched to other clusters while a low or negative value imply possible miss-classification. The overall silhouette score is obtained by averaging over the scores for each individual element and in our case we do a weighted average based on the total SRR of each element, so that cells with a high density have a bigger influence on the overall score. The resulting silhouette

scores for 2 to 24 clusters are shown in Fig. 4. We select the number of clusters based on three considerations: firstly, and primarily, by identifying localised maximums in the silhouette score, secondly by considering the applicability to our scientific question and lastly by accounting for practical aspects (e.g. very large number of clusters are not easy to analyse and visualise).

Based on the silhouette score (Fig. 4), we first select 6 clusters as this is a clear local maximum and a suitable number to identify the direction from which air masses approach CHC. In addition it is easy to analyse and interpret. However, this likely

lacks details and in our case, we are interested in the air mass footprint not only in terms of their direction from CHC but also the distance from the station and the vertical heights. Therefore we are also interested in a solution with more descriptive clusters. Therefore, in addition to $k = 6$, we also chose the next local maximum in the silhouette score which is $k = 18$ clusters. The 18 cluster solution adequately describes the data while maintaining a straightforward interpretation. Localised maximums in the silhouette score also exist for $k = 21$ and $k = 23$ clusters. However, we perform our detailed analysis using 18 clusters

rather than 21 or 23 clusters as it is more practical to work with a smaller number of clusters. To ensure the 6 clusters can be directly related to the set of 18 clusters (i.e. share the same boundaries), we first create the 18 clusters and then perform a

second round of clustering (starting from the 18 clusters) to obtain the 6 clusters. The 6 clusters are subsequently referred to as the 6 main pathways following the terminology used by Fleming et al. (2012).

## 4  Additional diagnostics and data

In order to understand and categorize air masses that have been in contact with the surface (where the emissions occur) we define the surface SRR percentage influence:

$$S_{\text{SRR}} = \frac{\text{SRR}_{\text{surface}}}{\text{SRR}_{\text{total}}} \times 100 \tag{2}$$

where the $\text{SRR}_{\text{total}}$ is equal to the theoretical total residence time of the simulation expressed in seconds ($4\,\text{days} = 345600\,\text{seconds}$).

In addition to air masses which have been influenced by the surface, we are also interested free tropospheric, ("clean") air
masses which are those that originate and remain above the boundary layer (BL). For this reason we also calculate the pseudo BL influence:

$$\text{PBL}^*_{\text{SRR}} = \frac{\text{SRR}_{<1.5\text{km}}}{\text{SRR}_{\text{total}}} \times 100 \tag{3}$$

where

$$\text{SRR}_{<1.5\text{km}} = \sum_{z=0}^{1.5\text{km}} \text{SRR}_{\theta rzt}. \tag{4}$$

Using the pseudo BL influence allows us to calculate the free tropospheric influence simply as $\text{FT}_{\text{SRR}} = 1 - \text{PBL}^*_{\text{SRR}}$. Thus the percentage of air not influenced by the lowest 1.5km is assumed to represent the free troposphere (FT). A pseudo boundary layer with a constant depth of 1.5 km also means that we neglect the diurnal variation in the PBL height. We make this approximation as due to our computational procedure, the specific depth of the PBL is lost when transforming the SRR output into log-polar coordinates. The choice of 1.5 km is motivated by previous studies that have quantified PBL depth in nearby
regions. For example, Carneiro and Fisch (2020) analysed radiosonde and remote sensing data from the GoAmazon project (Martin et al., 2016) and show that the typical minimum PBL height is 250 m and the deepest PBLs occur during daytime in the dry season and are 1.5 km deep on average. The global study by von Engeln and Teixeira (2013), based on reanalysis data, shows that PBL heights are somewhat deeper near CHC than in the Amazon and typically range between 500 m and 1.5 km. Therefore, the real PBL will usually be similar in depth or shallower than our value of 1.5 km. This means using the pseudo
BL depth will likely over-estimate the influence of PBL air masses and underestimate the FT influence.

When referring to the SRR percentage influence of a cluster, for simplicity we use

$$\text{SRR}[\%] = \frac{\text{SRR}}{\text{SRR}_{\text{total}}} \times 100. \tag{5}$$

This means that for every time step of the FLEXPART simulation, the sum of all the SRR cluster values adds up to $\text{SRR}_{\text{total}}$. In practice this theoretical value is not always achieved since inevitably a very small fraction of the particles leave the outer

domain (D01) before the end of the 4-day simulation. This also implies that for some time periods, the sum of all the cluster might not add up to 100%.

In order to describe the land-use characteristics of the geographical areas that the resulting clusters and main pathways originate from, we make use of the World Wildlife Fund (WWF) terrestrial ecoregions (TER, Dinerstein et al., 2017), WWF marine ecoregions (MER, Spalding et al., 2007) classification scheme, and two extra regions (EXR), namely the Lake Titicaca and the Southern Atlantic ocean regions, that we add to fulfil the domain. In total, 37 different ecoregions exist in the area covered by the largest WRF model domain. The advantage of using these regions is that they are well defined and described in the literature, do not depend on arbitrary political borders and are defined considering regions that have similar ecosystems and therefore similar potential emission signatures. Furthermore, the regions are also nested within biomes that provide a more general picture. Within our outermost domain (D01), 13 different biomes are present.

Sulfate mass concentrations measured at the CHC station from March to May 2018 are used in this study to illustrate an example application of our clustering methodology. The aim of this example is to show that our new method can identify the main source of the high sulfate concentrations measured at CHC. A very likely source of this is degassing from nearby volcanoes which was observed at the same time. The sulfate dataset is obtained using the quadrupole aerosol chemical speciation monitor (Q-ACSM, Aerodyne Research Inc.) which is able to routinely characterize non-refractory submicron aerosol species such as organics, nitrate, sulfate, ammonium, and chloride (Ng et al., 2011). Because of the low atmospheric pressure, a 130 $\mu$m diameter critical orifice was used in order to retain the normal sample mass flow rate (Fröhlich et al., 2013). In addition, inlet flow and mass calibration (using ammonium nitrate and ammonium sulfate as standards) were accomplished to guarantee optimal instrumental performance and mass quantification. The instrument's time resolution was 30 minutes. This sulfate timeseries was then correlated to the SRR timeseries of both the 18 clusters and the 6 main pathways.

# 5 Results

We now present the results. First in section 5.1 we present an overview of the mean SRR horizontal distribution. In section 5.2 we quantify the contribution of the surface, pseudo PBL, and FT sources to the air masses measured at CHC. In section 5.3 the characteristics of the 6 main pathways are presented before the more detailed analysis of the 18 clusters is shown in in section 5.4. Finally, in section 5.5 we show one example of how our clustering results can be combined with measurements to identify source areas.

## 5.1 Mean SRR spatial distribution

Before the clustering results are presented, we first give a brief overview of the 6-month average, vertically integrated SRR on the log-polar grid and its spatial distribution (Fig. 3). The average SRR is not uniformly distributed, even when similar horizontal ranges are considered. Two distinct large-scale areas exist with high SRR values. The first of these is the lowlands of Bolivia to the south-east and east of CHC and the second is the lowlands of Peru to the north-west as well as parts of the Pacific. These two distinct areas are divided by the Andes (see Fig. 2b) which run approximately north-south and act as a barrier. The

presence of the steep topography is also why areas of low SRR are identified in northern Bolivia; the easterly winds in this region are blocked and deflected by the topographic barrier preventing air masses from these region easily reaching CHC.

The inset in Fig. 3b also shows the average influence that La Paz / El Alto has on CHC. The average, vertically integrated
SRR values in this area are surprisingly low and much lower than other areas at a similar radial distance from CHC ( 20km). For example, the region 20 km north-east of CHC has average, vertically integrated SRR values that are more than double those found in the La Paz / El Alto area. Figure 3b only gives a very basic overview of where air masses that affect CHC originate from. At this point, before having clustered the SRR, the vertical distribution and temporal evolution of air masses influencing CHC cannot easily be determined nor visualised.

## 5.2   Relative contribution of the surface, PBL and FT to air sampled at CHC

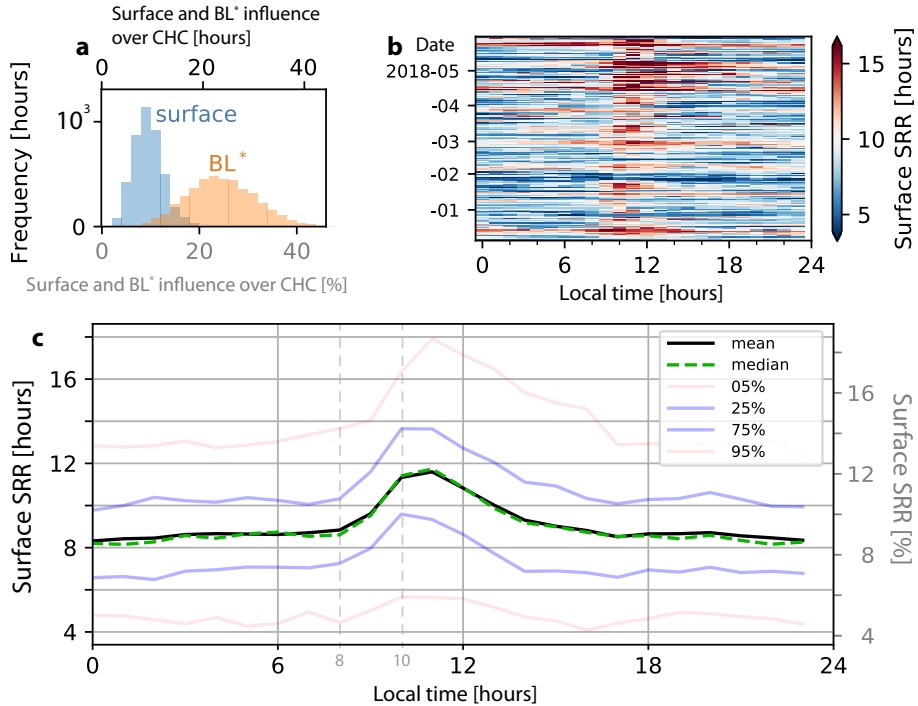

**Figure 5.** Panel a) is a frequency histogram showing the percentage (and residence time in hours—upper x-axis) of air sampled at CHC during each modelled hour that was in contact with the surface ($S_{SRR}$, Eq. 2, blue) or located within the pseudo PBL (PBL$_{SRR}^{*}$, Eq. 3, orange). Panel b) shows the surface influence (as a residence time) for every day of the modelled period as a function of local time. Panel c) shows the diurnal mean, median, and the 5,25,75 and 95 percentile ranges of the SRR surface influence. The left y axis is the residence time and the right y axis is the percentage influence.

As stated in the introduction, high-altitude mountain-top sites are often used in an attempt to sample free tropospheric air. In this section, we use the SRR output from FLEXPART to determine the relative influence of the surface, PBL and FT air on the

total sampled air mass at CHC over the 6-month period and the diurnal pattern of this influence. Our method is probabilistic and, for each observation time, determines the percentage of the sampled air mass influenced by the PBL or FT. Figure 5a shows

frequency histograms of the percentage influence of the "surface" and of a pseudo PBL (see Eq. 2 and Eq. 3 for definitions) for each hour. There is a strong linear correlation between the surface and PBL influence (slope=2.38 and $r^2$=0.9) which implies that the 1.5km-deep pseudo boundary layer is in general within the well mixed boundary layer.

The median influence of the surface is 9% meaning that on average 9% of the air sampled at CHC has been in contact with the surface in the last 4 days. In terms of the pseudo PBL, on average 24% of the sampled air masses represent PBL air. Indirectly

this means that approximately 76% of the air sampled at CHC can be considered representative of the FT. Note that this does not mean 76% of observation times are representative of the FT; it should be interpreted as, on an average simulation hour 24% of the measured air mass represents the PBL and the remainder the FT. This is a key strength of our method; it can determine at any given time what percentage of the sampled air mass arrives from different locations. An additional interpretation of the results shown in Fig. 5a is to consider the percentage of time when there is no influence (0 % on the x-axis) from the surface

or pseudo BL i.e. where there is purely free tropospheric air masses. This situation is never detected which indicates CHC is rarely representative of purely FT air. However, this is partly an artefact of the method employed here as all particles are forced to arrive at the surface (10 m a.g.l.) at the station.

Figure 5b shows the diurnal cycle of the surface influence for each day of the 6-month study period whereas Fig. 5c shows the average diurnal cycle. The largest positive values, indicative of a large surface influence, occur during daytime. The peak

emerges at 10 am local time (c) and happens almost every day (b). The duration of the high surface influence increases throughout the campaign with higher values of surface influence extending later in the afternoon in April and May. This gradual increase in the surface influence during the campaign might be explained by the transition towards the dry season when clear-sky conditions become more frequent increasing insolation periods which, in turn, favour deep well mixed PBL structures.

## 370   5.3   The six main pathways

In this section we describe the results from clustering the SRR log-polar cells into 6 groups. We call these 6 groups the main pathways (PW) since they tend to start near the station and reach far away from it as opposed to the 18 clusters that occupy more localised regions. Also, the limits of many of these are delimited by the Andean plateau. Furthermore, we label each cluster based on their 'clock direction' from CHC and append them with the acronym PW to distinguish them from the 18

clusters. For example, cluster label '03_PW' refers to the cluster whose centroid position is located east from the station.

In Fig. 6a and b we show the 6 main pathways (PW) along with the 18 cluster centroids. The pathways are the shaded coloured regions and contain 2 to 4 clusters from the 18-cluster grouping. The 03_PW is located geographically in the lowlands to the east of the station occupying the biomes 1, 2, 6, and 9 which in general are tropical forests and grasslands (Fig. 6e and f).

Cluster 05_PW originates from the south of the station in the altiplanic (montane grass and shrubland, biome 4) and lowland

(subtropical dry broadleaf, biome 6) regions. Horizontally, it follows the Altiplano plateau and its eastern slopes to the station. The cluster 07_PW comes from the south-west and most of its area is located above the Pacific Ocean/coast (biomes 3, 5, 7

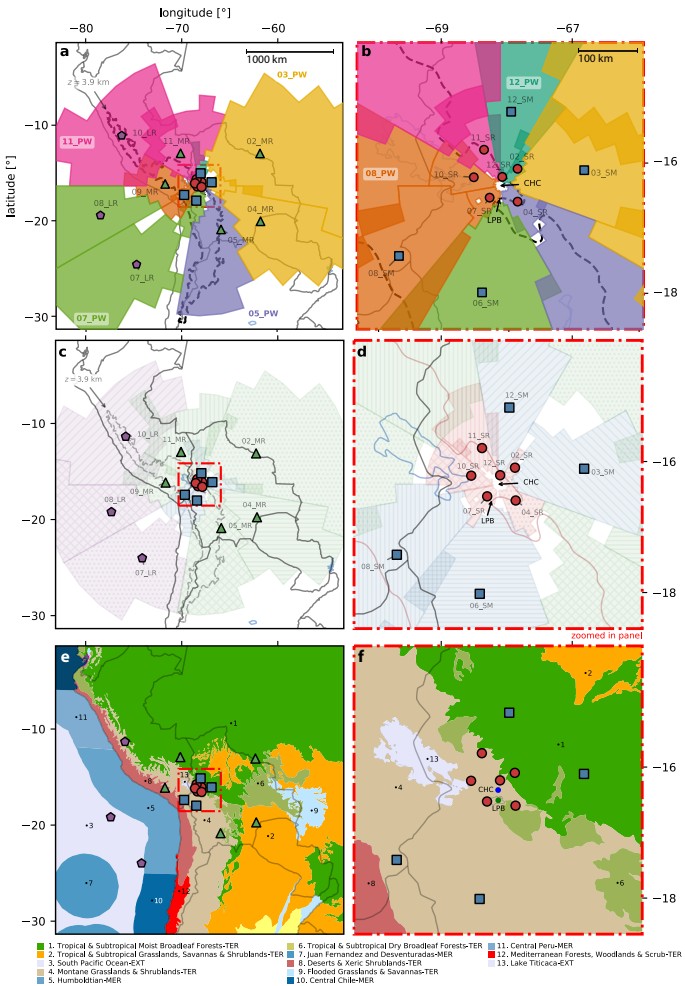

**Figure 6.** The shading in panels a) and b) show the horizontal extent of the 6 main pathways (PW). For easier visualization we only shade grid cells which have SRR values in the top 80% for each cluster. Each cluster centroid is marked with a disk (short range), square (short-medium range), triangle (medium range), or pentagon (long range) locator. The colour of each region is related to the main pathway they belong to: 03_PW (yellow), 05_PW (purple), 07_PW (green), 08_PW (orange), 11_PW (pink), and 12_PW (teal). The dashed black line corresponds to a height of 3.9 km a.s.l. and encircles the Altiplano plateau. Panel b) corresponds to the region inside the red rectangle in panel a). Panels c) and d) show the horizontal location of the 18 clusters. The colour is related to their distance range from CHC and the hatch distinguishes each cluster independently. Panel d) corresponds to the region inside the red rectangle in panel c). The colours, both for the centroid markers and the shaded areas, differentiate the distance range from CHC (SR, SM, MR, LR). The hatch patterns distinguish the area of each of the 18 clusters. Panels e) and f) show the biomes as described in section 4 and are shown to compare the area of the clusters and pathways to the underlying biome. The resulting area overlaps between each cluster and biome are shown in Fig. 9. Also a similar figure but showing the eco-regions rather than the biomes is shown in Fig. S3.

and 8). Cluster 09_PW comes from a relatively short distance and it occupies the altiplanic biome 4, the Titicaca lake (13) and two Pacific biomes (5, 8). The cluster 11_PW comes from the north and north-west. It occupies the lowlands, Altiplano and the Pacific coast i.e. biomes 1, 4, 11. As it gets closer to CHC, cluster 11_PW is located higher than 12_PW and thus goes above 12_PW. Finally, 12_PW comes from the lowlands north of the station and is contained within biome 1 (tropical broadleaf forest). Additional information containing the pathways boundaries and their spatial SRR distribution at different z-levels is shown in Fig. S9.

In Fig. 7a to d, we respectively describe the pathways' centroids in terms of their height above ground; height above sea level; surface influence ($S_{SRR}$, Eq. 2); and SRR percentage influence. In general, the farther the centroid is from CHC, the higher above ground level its centroid location is. The same pattern is observed for their height above sea level, however, if the location of the centroid is not too far away and above a location where the ground height is considerably lower than CHC, then the centroid location is located below the linear trend, e.g. 03_PW and 12_PW. In terms of their $S_{SRR}$, a decreasing trend is observed, in other words, the farther the centroid, the lower the influence from the surface. Therefore, 12_PW is highly influenced by its contact with ground (62 %, Figs. 7 and S2) while 07_PW is almost unperturbed by the surface (8 %). Finally, the SRR influence of each pathway seems to be uncorrelated with the distance from CHC. Pathway 03_PW has the highest influence over CHC with a share of 29 %. This is in agreement with previous studies at CHC (Chauvigné et al., 2019) where air masses from the Amazon were identified as the major contributors during the wet season DJFM (our modelling period covers DJFMAM).

Finally, in Fig. 8, we show the temporal influence of each of the pathways. We quantify the influence in percentage of each pathway by dividing the pathway's SRR values by the theoretical total residence time of the simulation, 96 hours × 3600 seconds. Note again that the sum of the influence for all 6 pathways shown in Fig. 8 does not always sum to exactly 100 which is due to particles leaving the domain. A clear change in the influence pattern at the beginning of May is seen: On one hand, the influence of pathways 03_PW and 05_PW and to a lesser extent 12_PW become almost negligible. On the other hand, pathways 07_PW and 08_PW increase their influence. This is consistent with Chauvigné et al. (2019) where it was shown that during the wet season (DJFMA) air masses from the lowlands and the east-southeast tend to have a bigger influence on CHC. In our case, 12_PW and 03_PW are clearly lowland pathways while 05_PW has a mixture of Altiplano and lowland influence that, nonetheless, comes from the southwest and therefore is mostly favoured during the wet season. The pathway 11_PW does not present a clear change during the 6 month period. Finally, visual inspection shows that the influence of each pathway varies on a timescale of 1-to-2 weeks. We will further develop this point when focusing on the detailed 18 clusters (section 5.4).

## 5.4 The 18 clusters

In the previous section, we provided a general picture of the air masses that influence CHC by using the 6 pathways. However, for an in-depth description, we now focus on the more detailed 18 clusters. We have subdivided these 18 clusters into 4 subgroups based on their horizontal distance from CHC: short range (SR), short-medium range (SM), medium range (MR) and long range (LR) for distances ranging from 0 to 100 km, 100 to 300 km, 300 to 800 km and >800 km respectively. Furthermore, we have labelled each cluster based on their distance range along with their 'clock direction' from CHC. For example, cluster

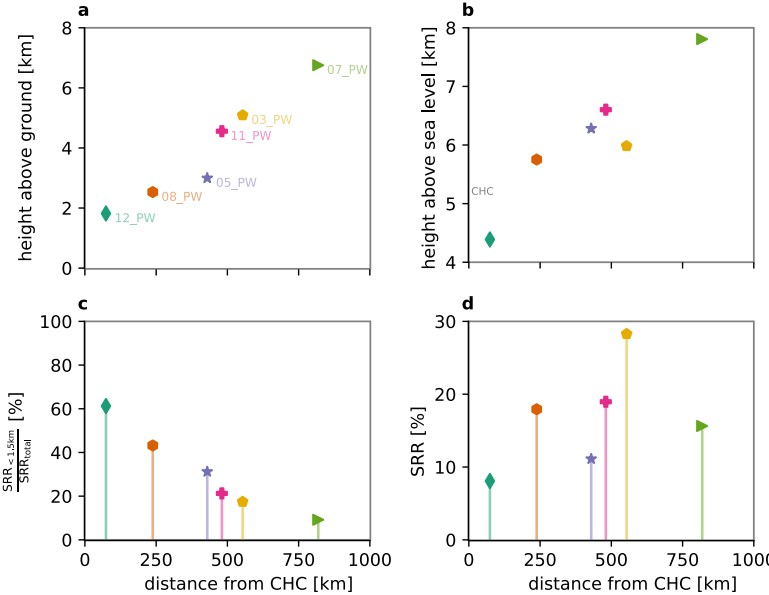

**Figure 7.** Centroid properties for each of the main pathways (PW). Panel a) shows the median height above ground level of each cluster while panel b) shows the median height above sea level. Panel c) shows the ratio between the SRR values that are below 1.5 km above ground level and the total SRR value for each cluster. Panel d) shows the mean SRR percentage for each PW. Quantitative numbers are presented in Fig. S2.

label '09_MR' refers to the cluster whose centroid position is located west from the station at a distance between 300 and 800 km. As done before, we first describe the clusters in terms of their horizontal location, then their vertical centroid properties and finally their temporal evolution. In Fig. 6c and d, we show the clusters geographical location along with a reference to the biomes that they occupy in space (Fig. 6e and f). Additional information containing the clusters boundaries and their spatial SRR distribution at different z-levels is shown in Fig. S1.

To further quantify the superposition of clusters and biomes presented in Fig. 6e and f, in Fig. 9 we show a heat map of the percentage of area that each cluster and biome share. For example, the geographical area of cluster 02_MR is split between the Tropical & Subtropical Moist Broadleaf Forests (biome 1, 56%), the Tropical & Subtropical Grasslands, Savannas & Shrublands (biome 2, 28%), the Tropical & Subtropical Dry Broadleaf Forests (biome 6, 10%), and the Flooded Grasslands & Savannas (biome 9, 6%). In general, there is a clear split between clusters located north-east (i.e. clock direction of 11 to 05) and southwest (clock direction of 05 to 11) from the station. The first are located in the generally more tropical and humid lowland biomes (1, 2, 6, 9) while the latter are located in drier altiplanic biomes (4, 13) and pacific biomes (3, 5, 7, 8, 10, 11, 8).

In Fig. 10a to d, we respectively describe the clusters' centroids in terms of their height above ground; height above sea level; surface influence ($S_{SRR}$, Eq. 2); and SRR percentage influence. In general, they follow the same patterns that we described for

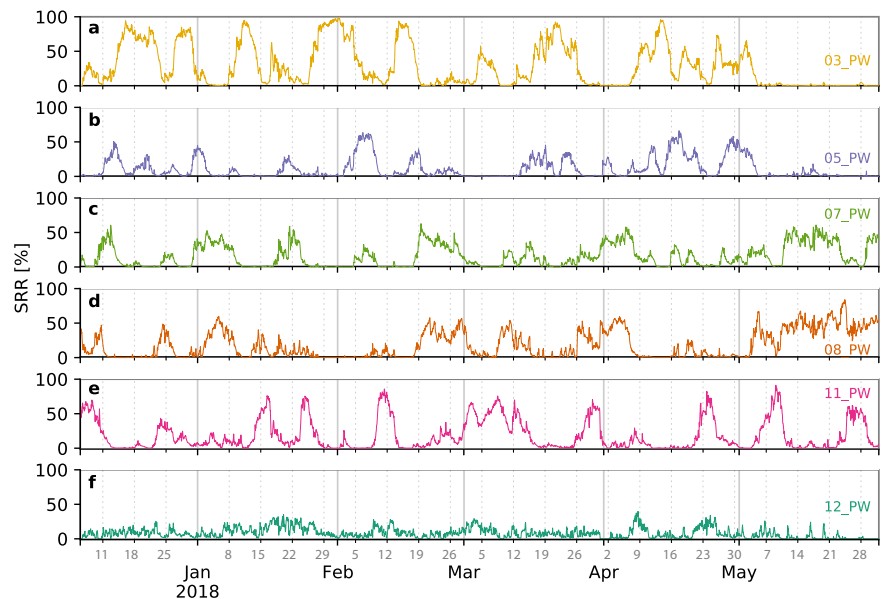

**Figure 8.** Time series of the SRR [%] influence for each of the 6 main pathways (PW) from 2017-12-06 until 2018-05-31.

| Biome | 02_MR | 02_SR | 03_SM | 04_MR | 04_SR | 05_MR | 06_SM | 07_LR | 07_SR | 08_LR | 08_SM | 09_MR | 10_LR | 10_SR | 11_MR | 11_SR | 12_SM | 12_SR |
|---|---|---|---|---|---|---|---|---|---|---|---|---|---|---|---|---|---|---|
| 1. Tropical & Subtropical Moist Broadleaf Forests-TER | 56 | 84 | 90 | 17 | 5 | 8 |  |  |  |  |  | 8 | 46 |  | 90 | 44 | 89 |  |
| 2. Tropical & Subtropical Grasslands, Savannas & Shrublands-TER | 28 |  | 6 | 66 |  | 19 |  |  |  |  |  |  |  |  | 2 |  | 10 |  |
| 3. South Pacific Ocean-EXT |  |  |  |  |  |  | 34 |  |  | 78 |  |  | 12 |  |  |  |  |  |
| 4. Montane Grasslands & Shrublands-TER |  | 16 |  |  | 26 | 62 | 65 | 100 |  |  | 35 | 29 | 6 | 65 | 8 | 35 |  | 100 |
| 5. Humboldtian-MER |  |  |  |  |  |  |  |  | 1 | 19 | 19 | 34 | 32 | 16 |  |  |  |  |
| 6. Tropical & Subtropical Dry Broadleaf Forests-TER | 10 |  | 3 | 10 | 69 | 5 |  |  |  |  |  |  |  |  | 2 | 21 |  |  |
| 7. Juan Fernandez and Desventuradas-MER |  |  |  |  |  |  |  |  | 25 | 3 |  |  |  |  |  |  |  |  |
| 8. Deserts & Xeric Shrublands-TER |  |  |  |  |  |  |  |  | 3 | 34 | 2 | 31 | 29 | 8 |  |  |  |  |
| 9. Flooded Grasslands & Savannas-TER | 6 |  |  | 7 |  |  |  |  |  |  |  |  |  |  |  |  |  |  |
| 10. Central Chile-MER |  |  |  |  |  |  |  |  | 16 |  |  |  |  |  |  |  |  |  |
| 11. Central Peru-MER |  |  |  |  |  |  |  |  |  |  |  |  | 11 |  |  |  |  |  |
| 12. Mediterranean Forests, Woodlands & Scrub-TER |  |  |  |  |  |  |  |  | 3 | 4 |  |  |  |  |  |  |  |  |
| 13. Lake Titicaca-EXT |  |  |  |  |  |  |  |  |  |  |  |  | 1 | 33 |  |  |  |  |

**Figure 9.** Heat map showing the biome and land cover characteristics associated with each of the 18 clusters. The percentages values indicate the percentage of air in each cluster that travels over each different biome. The darker the colour, the stronger the influence from the corresponding biome. For a description of the biomes see section 4. Due to rounding errors and the use of integers, some columns add to 99 rather than 100.

the 6 pathways with the exception that there are more clusters below the CHC station's height a.s.l. (Fig. 10b). These clusters, namely 02_SR, 12_SM and 03_SM are located close to CHC in the lowlands located north and north-east from the station.

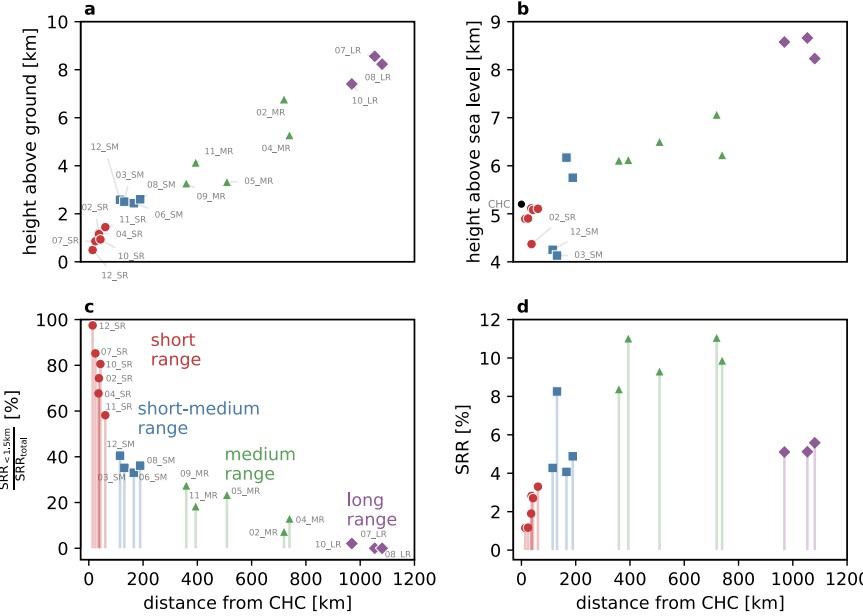

**Figure 10.** Cluster-median properties for each cluster. The horizontal axis for all panels represents the radial distance from CHC. Panel a) shows the median height above ground level of each cluster while panel b) shows the height above sea level. Panel c) shows the median ratio between the SRR values that are below 1.5 km above ground level and the total SRR value for each cluster. Panel d) shows the median SRR percentage for each cluster.

Specifically, all of the short-range clusters' centroids are below 2 km a.g.l. and below 5.2 km a.s.l. (Fig. 10a and b) which is below the CHC station height (5.2 km a.s.l.). Furthermore, these clusters are in close contact with the surface since their $S_{\mathrm{SRR}}$

(Eq. 2) is greater than 50 % (Fig. 10c).

The short-medium range clusters' centroids are between 2.4 km and 2.6 km a.g.l. However, their height a.s.l. varies from 4.1 to 6.1 km. This is due to clusters coming from both the Altiplano to the southwest and the lowlands to the north-east of the station. In general, only one third of these clusters' air masses are below 1.5 km ($S_{\mathrm{SRR}}$) and thus these clusters include notably more influence from the free troposphere than the SR clusters.

The medium-range centroids are between 3.2 km and 6.8 km a.g.l. This variance is mostly proportional to the distance from CHC (Fig. 10a) with clusters 04_MR and 05_MR being slightly below the rest due to their location in the lowlands. Their height above sea level varies from 6.1 to 7.1 km (Fig. 10b). In terms of their $S_{\mathrm{SRR}}$, these clusters vary from 7 to 20 %. These values are approximately inversely proportional to their centroid distance from the station (Fig. 10c).

Finally, for the long-range subgroup, their clusters' centroid far distance from CHC is reflected in their mean height a.g.l.

of 8.0 km, mean height a.s.l. of 8.4 km, and their low mean $S_{\mathrm{SRR}}$ of 0.7 %. Furthermore, due to this high altitude and low influence from the surface, air masses arriving from these clusters are likely to present free troposphere characteristics. These clusters are all located west of the station (clock direction 07-10).

The temporal evolution of the clusters is shown in Figs. 11 and 12. All clusters within the short-range subgroup show a high degree of temporal variability (Fig. 11a and b). Cluster 02_SR presents a high frequency variability which upon further analysis

(via Fourier transform, Fig. S4) is shown to be a clear diurnal pattern. However, this pattern does not happen everyday (Fig. S5) but in 81 out of the 176 modelled days (46 %). During these days, the peak happens in the early afternoon (13h local time). The

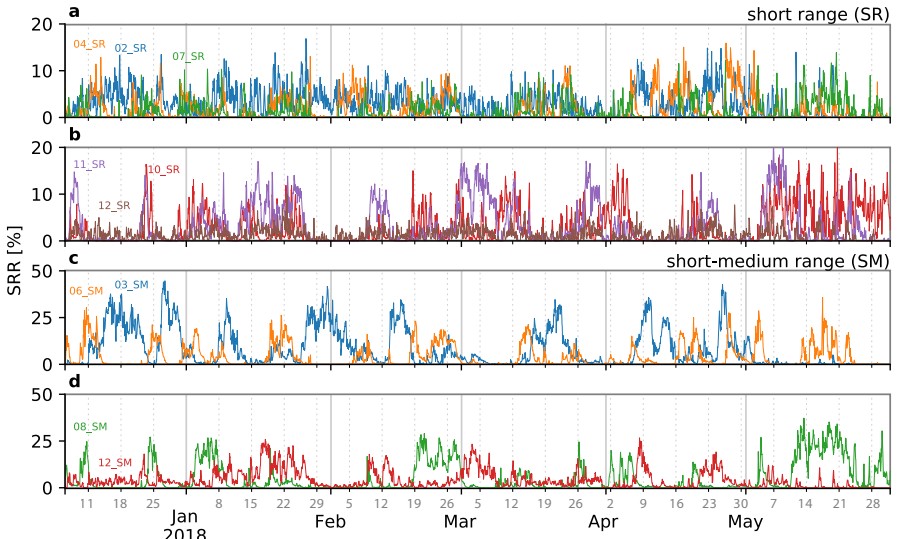

**Figure 11.** Time series of the SRR [%] cluster influence from 2017-12-06 until 2018-05-31. Panels a) and b) show the short-range (SR) clusters, c) and d) show the short-medium-range (SM) clusters. Note the different y-axis between panels a), b) and c), d).

cluster 04_SR is highly variable as well however it does not present a diurnal pattern (Fig. S4) and its variation is similar to that of 05_MR, probably due to the fact that both clusters belong to the same pathway. Cluster 07_SR does present a sharp diurnal pattern in 41 (23 %) out of 176 modelled days (Fig. S6) peaking at 11h local time. We conjecture that the peak time difference

between 02_SR (13h) and 07_SR (11h) is due to the different land type; 02_SR originates from the high humidity biome 1 (Tropical and subtropical forest, Fig. 6f) whereas 07_SR originates from the less humid biome 4 (Montane grasslands). This difference would entail different thermal inertia, different diurnal cycles and partitioning of the sensible and latent heat fluxes and thus a different boundary layer evolution. Cluster 07_SR is of particular interest since intense anthropogenic emission sampled at CHC would most likely be generated in this highly populated area. Furthermore, the close contact of cluster 07_SR

with the surface and its diurnal variability favour transport of emissions to the station during the day when PBL air from La Paz and El Alto is advected upslope by thermally driven winds. The cluster 10_SR, which originates close to Lake Titicaca, presents a diurnal pattern in 63 (36 %) out of 176 days peaking at 8h local time (Fig. S7). We conjecture that this early morning peak is related to the lake breeze circulation that develops due to the temperature difference between land and the lake. The average surface temperature of the lake, obtained from Pillco Zolá et al. (2019), is 10° C and does not have a diurnal cycle.

At 8h local time, the lake's surface temperature is higher than the surroundings favouring a land breeze (airflow from land to the lake) near the surface and ascent over the lake. The return flow, near the top of the BL, potentially advects air masses from

the lake to CHC. The cluster 11_SR does not present a diurnal variation (Fig. S4) and its influence seems to be mostly driven

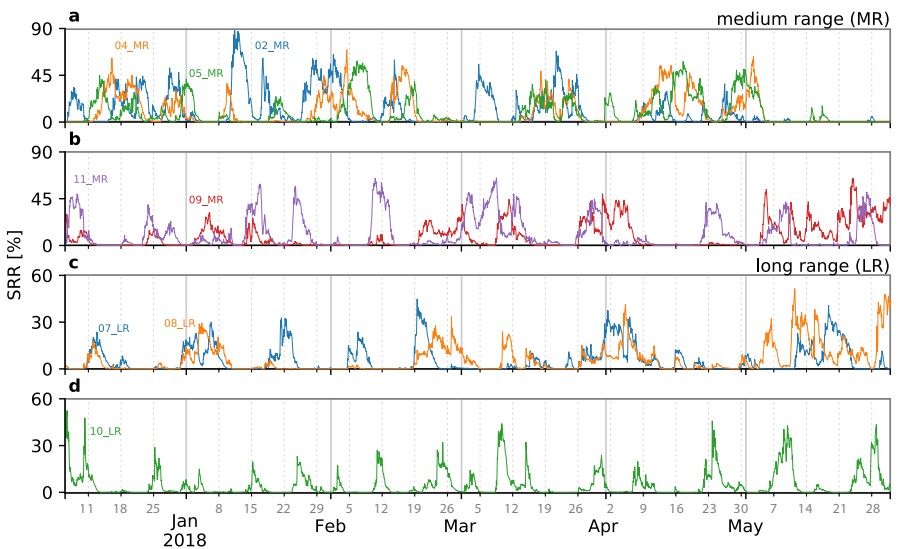

**Figure 12.** Time series of the SRR [%] cluster influence (similar to Fig. 11). Panels a) and b) show the medium range (MR) clusters. Panels c) and d) display the long range (LR) clusters.

by the medium-range cluster 11_MR (Fig. 12). Cluster 12_SR does present a diurnal pattern (Fig. S8) in 48 days out of the 176 modelled days (27 %) with a peak around 10h local time. This earlier peak, in comparison to 07_SR, is most likely due
to its close location to CHC (the closest one, 14 km ) so that early morning irradiation would create favourable conditions for up-slope winds.

The short-medium (SM), medium (MR) and long (LR) range clusters' temporal variability is shown in Fig. 11c and d, and Fig. 12a to d. The power spectra of the SRR intensity for these clusters is shown in Fig. S4 and is between 1 and 2 weeks except for 08_SM and 12_SM which, in addition, also show a small diurnal variability.

The prevalence of some short-medium range clusters changes during the 6-month period. Clusters 03_SM, 12_SM, 02_MR, 04_MR and 05_MR occur regularly from December (wet season) to April (transition season) but cease influencing in May (dry season). In contrast, the influence of clusters 06_SM, 11_MR and 10_LR do not change substantially during the 6 months whereas the influence of clusters 08_SM increases 09_MR, 07_LR and 08_LR in May.

Finally, in Fig. 13 we present a quantitative summary of the 18 clusters centroid properties along with their average SRR
influence. Furthermore, we also link each of the 18 clusters to their main pathway. On average, there are 3 clusters per pathway and at least two distinct distance ranges. Furthermore, the clusters within each pathway are heterogeneous in position, surface influence and age which supports the idea that further insight into the air mass transport patterns can be attained by further subdividing the pathways.

| short name | SRR [%] $n_c=18$ | distance from CHC [km] | height above ground [km] | height above sea level [km] | $\frac{SRR_{<1.5km}}{SRR_{tot}}$ [%] | age [h] | main pathway | SRR [%] $n_c=6$ |
|---|---|---|---|---|---|---|---|---|
| 03_SM | 8.5 | 134 | 2.5 | 4.1 | 36 | 34 | 03_PW | 29.0 |
| 02_MR | 10.8 | 671 | 6.8 | 7.1 | 7 | 60 | | |
| 04_MR | 9.6 | 687 | 5.2 | 6.2 | 14 | 58 | | |
| 04_SR | 1.9 | 36 | 1.2 | 5.1 | 70 | 18 | 05_PW | 11.4 |
| 05_MR | 9.6 | 506 | 3.3 | 6.5 | 24 | 50 | | |
| 06_SM | 4.3 | 169 | 2.4 | 6.1 | 34 | 35 | 07_PW | 13.9 |
| 07_LR | 4.7 | 972 | 8.5 | 8.6 | 0 | 64 | | |
| 08_LR | 4.9 | 962 | 8.1 | 8.1 | 0 | 61 | | |
| 07_SR | 1.2 | 25 | 0.8 | 4.9 | 87 | 12 | 08_PW | 17.7 |
| 10_SR | 2.8 | 43 | 0.9 | 5.0 | 82 | 14 | | |
| 08_SM | 5.0 | 188 | 2.5 | 5.7 | 38 | 37 | | |
| 09_MR | 8.8 | 355 | 3.2 | 6.1 | 29 | 45 | | |
| 11_SR | 3.3 | 61 | 1.4 | 5.1 | 60 | 27 | 11_PW | 19.6 |
| 11_MR | 11.4 | 390 | 4.2 | 6.1 | 19 | 54 | | |
| 10_LR | 4.9 | 917 | 7.4 | 8.6 | 2 | 68 | | |
| 12_SR | 1.2 | 14 | 0.5 | 4.9 | 98 | 9 | 12_PW | 8.4 |
| 02_SR | 2.9 | 36 | 1.2 | 4.3 | 77 | 15 | | |
| 12_SM | 4.3 | 119 | 2.6 | 4.2 | 42 | 41 | | |

**Figure 13.** Properties of the eighteen clusters ($n_c = 18$). The short name's digits refer to the clockwise direction of the centre of mass of each cluster. The letters refer to the range: SR = short range, SM = short-medium range, MR = medium range, and LR = long range. The SRR [%] column describes the average contribution of each cluster. We also show the distance from CHC, height above ground and height above sea level of each cluster's centre of mass. Furthermore, $\frac{SRR_{<1.5km}}{SRR_{total}}$ [%] shows the ratio between the SRR below 1.5 km and the SRR summed over the full vertical column ($SRR_{total}$). The age column shows the weighted median time (in hours) required by the air masses to arrive from the respective cluster to CHC. The last two columns describe the results of clustering the 18 clusters into 6 clusters (main pathways, $n_c = 6$). The digits also refer to the clockwise direction. The last column adds up the SRR [%] of the cluster belonging to each main pathway. The colours in the first column correspond to the 6 pathways and associated colours shown in Figs. 6a and b, 7, and 8.

## 5.5 An example application: sulfate from volcanic degassing

This section presents a proof of concept for our newly developed method to identify sources regions of air sampled at CHC. We use in situ observations of particulate sulfate at CHC taken with a Q-ACSM instrument, together with the results of our air mass history clustering analysis presented above to identify the source of the emissions. During March and April, satellite imagery showed that while the Ubinas volcano was not degassing, the Sabancaya volcano, located 400 km WNW from CHC, was emitting and thus there was a clear, known, almost point source of particulate sulfate. As there are no other comparable

strong point sources of particulate sulfate in the domain of interest, we assume that when high levels of particulate sulfate were measured at CHC, the air mass passed through the area near the volcano. Therefore, there should be a high correlation between the time series of sulfate and the SRR time series for clusters originating near the volcano.

To determine if this is the case, we calculated the 'Pearson' correlation coefficients for all available measurements of sulfate from the Q-ACSM to the SRR time series of each of the 18 clusters and all of the 6 pathways which are shown in Fig. 14a.

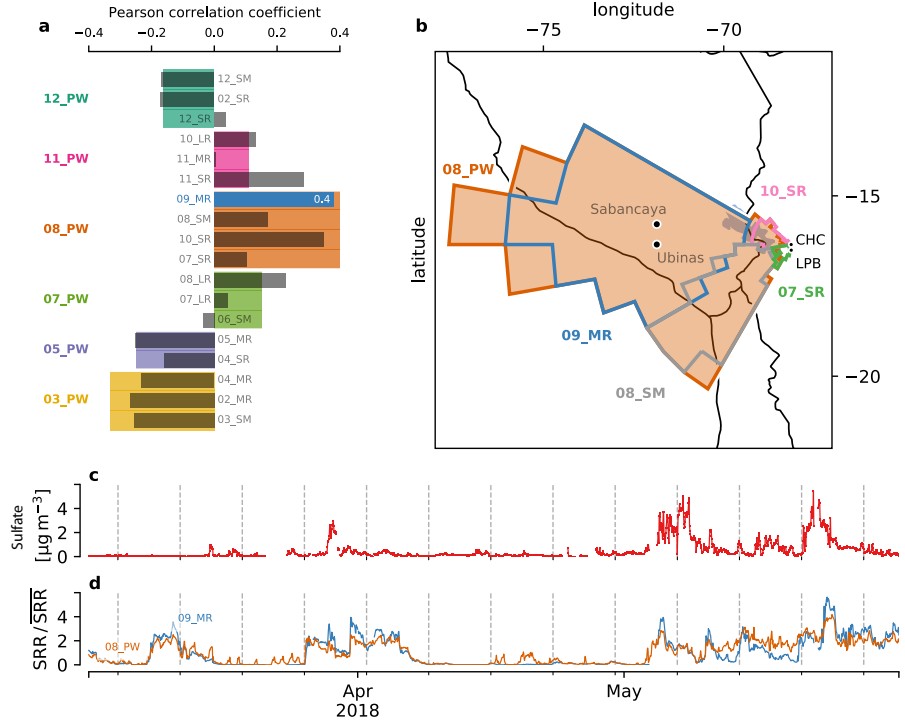

**Figure 14.** In panel a), we show the Pearson correlation coefficient between sulfate concentrations sampled at CHC and, both, the 18 clusters and the 6 pathways (PW). In panel b), the regions covered by the pathway 08_PW and the clusters 09_MR, 08_SM, 10_SR and 07_SR are shown along with Sabancaya and Ubinas volcanoes (both are located within 09_MR). All of the aforementioned clusters are contained by 08_PW. Panel c) and d show the timeseries of sulfate sampled at CHC and the normalized SRR timeseries of 08_PW and 09_MR respectively.

The cluster and pathway with the highest coefficients are 09_MR (0.40) and 08_PW (0.42) and both correlations have a $p_{value} < 0.001$. The horizontal locations of this cluster and pathway are shown in Fig. 14b and their corresponding timeseries are shown in Fig. 14d. The timeseries for the sulfate measurements is shown in Fig. 14c. The correlation in combination with either the simplified (6 pathways) and the specific (18 clusters) clustering scheme correctly assigns the source region to the location of the degassing Sabancaya volcano.

The coefficients from the pathways are quite clear; the highest value of 0.42 is at least twice as high as the next candidate, 07_PW. This clearly distinguishes 08_PW as the best candidate for the source of measured sulfate. The 18 clusters also assign the highest coefficient value correctly to 09_MR. However, 10_SR, 11_SR and 08_LR all have similarly high correlation coefficients. Therefore, while the results from 18 clusters are better at pinpointing the source region of sulfate, other regions also appear as plausible candidates so there is a risk of overfitting when too many regions are considered. This may be a result

of this source regions being entangled in terms of their temporal influence over the station. For example, in this particular case, the high correlation of 10_SR, 11_SR and 08_LR could be attributed to the fact that air masses that travel through the region of the volcano also have a time residence in the regions where 10_SR, 11_SR and 08_LR are located.

    This example shows that the clustering scheme is successful in identifying regions in its simplified version (6 pathways) and also in its more detailed version (18 clusters) albeit care must be taken when drawing conclusions so that we do not overfit to

the source regions. Finally, it should be noted that only one example is presented here. Future work will include the extension to additional case studies, for example, comparing the clusters to measurements of black carbon.

## 6   Discussion and recommendations

Previously, we described the characteristics and location of the 6 pathways and 18 clusters and related this to the surface type (biomes). It is also relevant to consider how the pathway positions and time series relate to typical meteorological patterns.

During the austral summer, the Intertropical Convergence Zone (ITCZ) migrates south, coinciding with the decrease of the meridional gradient temperature and the associated southward shift of the westerly subtropical jet stream. At the same time, deep convection starts to develop especially in the central part of the continent. This favours the expansion of the equatorial easterly winds and thus a weak mean east-to-west flow in the middle and upper troposphere is established (Garreaud et al., 2003). This east-to-west flow is well captured by 03_PW (03_SM, 02_MR and 04_MR) which is strong in DJFMA.

At the same time, the expansion of the trade winds also generates the South American Low Level Jet (SALLJ) which brings moisture from the Atlantic and the Amazon basin first to the eastern slopes of the Andes and then turning in a north-to-south fashion towards the southern part of the continent. It is known that the SALLJ is also responsible for bringing moisture to the Altiplano plateau since part of this flow is channelled into/goes over the plateau (Insel et al., 2010). This pattern correlates well with 12_PW (12_SR, 02_SR and 12_SM) which brings low altitude air masses from the eastern slopes of the station. The

pathway also is strong during DJFMA and diminishes its influence in May. It is important to note that 03_PW and 12_PW do not necessarily influence the station synchronously.

    During the austral winter, the ITCZ migrates north and the subtropical westerly jet stream moves north reaching up to 20 degrees south (Garreaud et al., 2003). This creates an upper-level, large-scale westerly flow that favours air masses from the Pacific/Altiplano region. This coincides with the increment of the influence of both 07_PW (06_SM, 07_LR and 08_LR) and

08_PW (specially 08_SM and 09_MR). The first brings long-range subsiding air masses from high up in the troposphere (7.7 km a.s.l.) while the second advects dry air from Altiplano at low-level (2.5 km a.g.l.). Both of these clusters are present during the dry season and reach maximum intensity in May.

    The connections of pathways 05 and 11 to the general atmospheric dynamics is not immediately evident. Pathway 05_PW comes from the south and is strongest between December and March and has a low centroid height a.g.l. (3 km). This could

be linked to the development of a strong South Atlantic Convergence Zone starting in the eastern slopes of the Altiplano at latitudes of around 20° that would pull surface air from the south. On the other hand, 11_PW which consists of 3 clusters,

originates from both from the Amazon and the Pacific (Fig. 9). These multiple sources areas, spanning both east and west, indicate that 11_PW is a "hybrid" pathway and therefore likely occurs in both the wet and dry seasons.

This study builds upon a previous source region analysis performed by Chauvigné et al. (2019) at the CHC station where a similar WRF setup was used in combination with back trajectories. Our methodological approach is notably different from this earlier study primarily as we use a Lagrangian dispersion transport model rather than a back trajectory model meaning that turbulent mixing and convection processes that air parcels experience during transport are better represented in this study. Additionally, our WRF simulations and hence meteorological data have a higher vertical resolution (61 levels compared to 28) and our 18 clusters provide more detail than the 6 clusters presented by Chauvigné et al. (2019). Our pathway results are largely in agreement with this previous study: similar source regions are observed for similar seasons and both studies show that source regions from the west start influencing the station in the transition month of May. However, key additions of this study are (1) the vertical distribution of the air masses sources is more accurately captured, (2) the influence of the surface and the pseudo PBL on air sampled at CHC is more accurately quantified and (3) the diurnal cycle is captured by the analysis.

The type of analysis performed here is applicable to many other stations worldwide both in mountainous regions but also for stations in non-mountainous areas which are equally influenced by local and remote sources. Therefore, based on the experience and knowledge gain here, we make the following recommendations for future studies:

- For source identification in regions with complex terrain we strongly recommend the use of Lagrangian dispersion models over simple, limited number back trajectories based approaches. This has been previously noted also by Stohl et al. (2002).

- The accuracy of the meteorological input data is crucial for reliable results and therefore should be verified before performing the FLEXPART simulations. In our case, this step revealed large biases in the Titicaca lake temperature which affected local wind patterns.

- Selecting the optimal number of clusters is challenging particularly in situations like this where there is continuum rather than a clear number of clusters. Consequently, in addition to quantitative scores, the scientific applications and practical aspects should be considered when selecting the number of clusters. We recommend that both pathways (smaller number) and clusters (larger number) are computed and their centroid characteristics analysed. Both are useful and have notably different applications.

- Due to computational limitations, and given that the campaign only lasted for six months, we only simulated six months. For a more complete physical understanding, if computational resources allow, we recommend that a full annual cycle is always simulated even if the observational campaigns the model simulations support are shorter in duration.

## 7 Summary and Conclusions

In this study we successfully developed a new method to identify air mass source regions in sites of complex topography. We then applied this methodology to the GAW station CHC, located near La Paz / El Alto at 5240 m.a.s.l. In order to accomplish

this, we started with a WRF simulation in combination with FLEXPART to create a high-resolution data set of source areas for CHC. Then we applied our new method, based on cluster analysis, to transform the complex and large output dataset into a user-friendly timeseries dataset of air mass source regions. We documented the characteristics of the identified source areas and demonstrated the strength and simplicity of the method's classification results by applying our method to confirm that the Sabancaya volcano is the source of sulfate measurements at the CHC station. The main conclusions of our analysis are:

– On average, 9% of the air sampled at CHC has been in contact with the surface, and 24% with the pseudo PBL, within the previous 4 days. Therefore we can conclude that on average, at any given time 76% of the measured air mass at CHC represents free tropospheric air. Thus, the air masses sampled at CHC are very rarely purely free tropospheric air masses (Fig. 5a).

– The surface influence has a clear diurnal cycle, with low contributions during the night and higher contributions starting at 10 am local time and continuing during the day. The duration of the high surface influence during daytime is longer in the dry season (May) compared to the wet season (December–March, Figs. 5b and c).

– Air masses arriving at CHC have a wide range of sources covering many different biomes and altitudes and it is common for any one specific sample time to have more than one source region (Figs. 6, 8, 11, 12 and S3).

– The most dominant pathway to emerge in our 6-month study is 03_PW which is responsible for 29% of the SRR and originates in the Amazon. However, as we detected that this PW does not occur in May, we hypothesize, based on Chauvigné et al. (2019), that if our analysis extended over all of the dry season (May–August), the overall prevalence of this PW would decrease and others (e.g. 07_PW and 08_PW) would increase (Figs. 7 and S2).

– For the clusters' centroid positions, a linear relationship exists between the horizontal distance from CHC and the height above ground, with those farther away also being located higher in the atmosphere (Figs. 7 and 10).

– Clusters located closest to CHC have the highest pseudo PBL influence and, rather than a linear decrease, the influence of the pseudo PBL decreases almost exponentially with increasing distance from CHC (Fig. 10).

– The contribution to the SRR is largest for the medium-range clusters and smallest for the short-range clusters thus showing no linear relationship with the distance from CHC (Fig. 10).

– The short-range clusters have high temporal frequency modulated by local meteorology driven by the diurnal cycle whereas the mid- and long-range clusters' variability occurs on timescales- governed by synoptic-scale dynamics (Figs. 11, 12, S4, S5, S6, S7 and S8 ).

To conclude, firstly, the method developed here can be applied to many other long term monitoring stations. Secondly, the data sets produced here, that provide detailed information about the sources of air masses sampled at CHC, will be applied in forthcoming studies on the chemical composition measurements made at CHC during the SALTENA campaign.

## Appendix A: Additional FLEXPART output pre-processing and clustering details

The purpose of this appendix is to give additional technical details concerning how the raw FLEXPART output was processed and subsequently clustered. In addition, more detailed justification for the choices made in this process are also given here.

### A1 Vertical ($dz$) FLEXPART output levels

The two nested output grids of FLEXPART were defined as described in section 2.2. The vertical grid was selected to have a constant $dz$ of 500 m instead of the customary varying resolution a.g.l. (usually the vertical resolution is higher close to the surface than aloft). However, as we are in an area of complex terrain the constant $dz$ was chosen so that comparison of vertical grids for locations with considerable different ground height a.s.l. is easier. For example consider the grid cells above CHC (5 km a.s.l.) and La Paz (3.6 km a.s.l.). If we were to use varying vertical resolution a.g.l. then an air mass moving along the same pressure level as CHC would move from a high resolution vertical grid to a low resolution vertical level in less than 20 km (the horizontal distance from CHC to La Paz). We selected a $dz$ of 500 m as a compromise; ideally we want as small as $dz$ as possible near the surface but to minimize computational cost we want a large as possible $dz$ and thus fewer vertical levels. The constant $dz$ also makes the conversion between a.s.l. and a.g.l. seamless.

### A2 Rectangular to log-polar regridding of the SRR matrix (step 3)

Since we are in the tropics, it is reasonable to use an equirectangular projection to a Cartesian coordinate system defined by longitude (lon), latitude (lat) and height above ground level ($z$). Any point (lat, lon, $z$) can be represented in polar cylindrical coordinates and is given by

$$r = \sqrt{(\text{lon} - \text{lon}_c)^2 + (\text{lat} - \text{lat}_c)^2}$$
$$\theta = \tan^{-1}\frac{\text{lon} - \text{lon}_c}{\text{lat} - \text{lat}_c}$$
$$z = z$$

where $r$ is the radial distance to the receptor location ($\text{lon}_c$, $\text{lat}_c$) and $\theta$ is the clockwise angle starting north from the receptor. Notice that $r$ is the Euclidean distance of lat and lon. The relation between $r$ and the geodesic distance $d$ in km is given by the approximation

$$d\,[\text{km}] = 108.6\frac{\text{km}}{°}\, r\,(\pm 3\%)$$

and is valid for the whole region covered by the WRF D01 domain. The radial boundaries of the log-polar grid are separated by a distance $\Delta\theta = 10°$. The radial length $\Delta r$ of the log-polar cells is $\Delta r = r_{i+1} - r_i$ where $r_{i+1} = r_i e^a$ and $e$ is Euler's number (2.71). The value of $a = 0.18$ is chosen so that the log-polar cells approximate a square with sides $\Delta r \approx r_i \cdot \Delta\theta$. The ring radii of the log-polar grid are determined by starting with a initial ring $r_0$ of radius $0.08°$ ($\approx [8.7]km$). The choice of the value for $r_0$ should be large enough to allow the first ring of radial cells to have an area larger than the grid cells in the highest resolution

output from FLEXPART so that at least one original FLEXPART output grid falls on each radial cell (with this configuration, the 550-km$^2$ urban area of La Paz / El Alto is covered by 37 log-polar grid cells). The following 30 ring radii are obtained iteratively using $r_{i+1} = r_i e^a$.

Once the new log-polar grid is defined, the SRR must be regridded from the original longitude-latitude grid to this new grid. For each log-polar grid cell, the SRR is obtained by adding the SRR values of the rectangular grid cells whose centre of mass is contained within the log-polar grid cell.

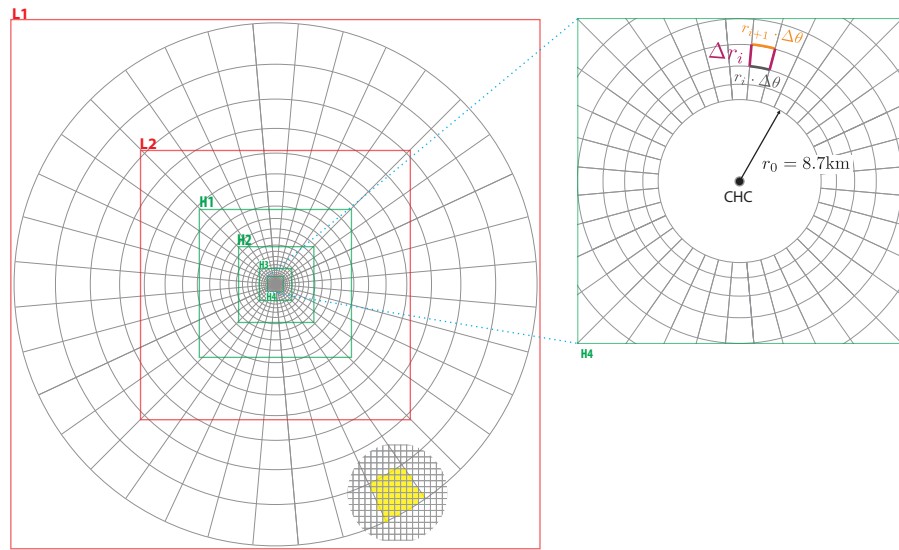

**Figure A1.** Schematic of the rectangular to log-polar regridding. See section A2 for a full description.

Given the considerable amount of data, the regridding procedure needs to be very computationally efficient. The straight-forward way to regrid the dataset would have been to find the volume that each rectangular grid shares with each log-polar grid cell and then distribute the SRR value accordingly. However this method proved to be computationally too demanding and an alternative computationally-efficient method with similar logic was devised and is shown schematically in Fig. A1. The alternative method is performed in a procedural manner starting with a log-polar grid cell in the outermost ring. The log-polar cell's SRR value is obtained by adding all the rectangular grid cells (yellow squares in Fig. A1) from the coarse resolution rectangular grid (L1) whose center fall within the log-polar grid cell. Then we proceed to obtain the SRR value of all other log-polar grid cells in the same ring and then continue with the next ring until we reach the inner most ring. As a rule of thumb at least 50 rectangular grid cells are required per one log-polar grid cell. If this condition is not met, then the rectangular grid resolution is increased by splitting each cell into 4 equal-sized rectangular cells with 1/4 of the original SRR value (for example that is the case for L2, H2, H3 and H4) or, if available, by choosing a rectangular grid with higher resolution (for example H1). In this manner we ensure that each log-polar grid cell is obtained from the most resolved rectangular grid available and in the case that the log-polar grid-cell size starts to be comparable to that of the collocated rectangular grid, then, the rectangular

grid is subdivided so that the SRR values of the rectangular grid cells are proportionally mapped onto the log-polar cells. The output of the regridding is a 4-dimensional array: $\mathbf{SRR}_{\theta rzt}$.

## A3 Pre-processing of the SRR matrix (step 4): smoothing, normalization and filtering

The $\mathbf{SRR}_{\theta rzt}$ matrix is smoothed in all 4 dimensions using scipy's Gaussian filter function (Virtanen et al., 2019). The standard deviations of the Gaussian filter are given for each dimension and are $t = 3, z = 0.25, r = 1, \theta = 0.5$. The purpose of smoothing the data is solely to improve the accuracy of the clustering. Clustering without smoothing produces very similar results except that when smoothing is not used, very low and intermittent influence cells are assigned clusters randomly rather than matching their neighbours. The smoothing forces these few 'problematic' cells to be assigned to the neighboring group. Once the clustering has been performed, and each grid cell allocated to a cluster, the subsequent analysis is performed using the non-smoothed $\mathbf{SRR}_{\theta rzt}$.

We are interested in grouping elements depending on their variation over time rather than their absolute values, therefore we need to normalize the dataset. We use scikit-learn's (Pedregosa et al., 2011) quantile transform function to normalize our elements to a uniform distribution. This procedure has the advantage of being robust to outliers and also performs quite well with sparse arrays or semi-sparse arrays like ours. After this function is applied, the distribution of each element resembles a uniform distribution with a value range from 0 to 1. In order to accomplish the normalization, the smoothed $\mathbf{SRR}_{\theta rzt}$ is then transformed from a 4-dimensional array to a two-dimensional array which as one dimension has the arrival time $t$ and as the second dimension has all of the grid cells ($p$): $\mathbf{SRR}_{pt}$. This transformation means that the clustering scheme will not know which cells are neighbours—hence the need to smooth $\mathbf{SRR}_{\theta rzt}$ while still in 4-dimensional space. The transformation step (a.k.a. stacking) into a 2 dimensional array is normal in k-means clustering as it is a requirement of the algorithm. Quantile mapping is then applied along the time dimension of the 2D array, $\mathbf{SRR}_{pt}$ using scikit-learn's quantile mapping function (QuantileTransformer, default options). The output of this step is a 2D array with the same dimensions as $\mathbf{SRR}_{pt}$, however instead of the SRR values, the values now range from 0 to 1. This is denoted $\mathbf{SRR}'_{pt}$.

The last step before clustering is to remove the grid points from $\mathbf{SRR}'_{pt}$ whose SRR values are either zero or have very little influence and including them adds computational burden to the clustering algorithm and does not improve the results. However, this also makes scientific sense—we do not want to include grid cells into any cluster if air from those grid cells never (or very rarely) arrives in CHC. In order to decide which elements are not beneficial based on the above definition, we define a threshold T in the following way. First, we sum all the SRR values for each of the elements over the time period. Then, we sort the element based on their total SRR value and compute the cumulative SRR values. Finally, we split this dataset at the point where 85 % of the total SRR value is reached and discard the remaining 15 %. This procedure leaves us with 8580 cells out of the total 33480 in the grid. Out of the excluded cells, 24.8 % had a zero SRR total value and the total median for the non-zero left out cells is 2060 s. The total median for the included 8580 cells is 112190 s. The output of this step is $\mathbf{SRR}'_{\rho t}$, where $\rho$ are the retained grid points.

## A4    The clustering algorithm

$\mathbf{SRR}'_{\rho t}$ is then used as input to the k-means clustering algorithm. The aim of k-means clustering is to minimize a distance metric between each cluster member and the cluster centroid. Mathematically this is achieved by minimizing the function

$$J = \sum_{j=1}^{k} \sum_{i=1}^{\rho} \|x_i - c_j\|^2 \tag{A1}$$

where $k$ is the number of clusters, $\rho$ is the number of grid points, $x_i$ is the $i^{th}$ grid cell and $c_j$ is the cluster centroid. $\|x_i - c_j\|^2$ is the Euclidean distance which is the distance metric used here.

The number of clusters, $k$ is first defined and their centroids positions (i.e. cluster centre points) are initially randomly specified. Each grid cell (element) is then assigned to the cluster to which it is closest to based on the Euclidean distance. Once each element is assigned to a cluster, new centroid positions are computed. An iterative procedure then takes place, with the elements re-assigned to clusters based on the newly computed centroids. This iteration continues until either convergence is achieved or the maximum number of iterations is completed. Convergence is determined by considering the residual of the sum

of each individual euclidean distance. The final output of the k-means clustering is that each grid cell (element) is assigned a cluster number.

*Data availability.*    –  The SRR cluster timeseries are publicly available at https://doi.org/10.5281/zenodo.4539590.

– Both the WRF and FLEXPART raw output datasets are available upon request.

– The source code for the FLEXPART-WRF model 3.3.2 used in this study can be downloaded from https://www.flexpart.eu/downloads.

We modified parts of the original code to adapt it to the complex topography of the domain. The modified source code can be downloaded from https://doi.org/10.5281/zenodo.5516295

– The source code for the WRF model version 4.0.3 used in this study can be downloaded from https://www2.mmm.ucar.edu/wrf/users/

*Author contributions.*  DA performed the WRF and FLEXPART simulations and did the data analysis. EK corrected the lake temperature problem and verified the WRF simulations. DA and VAS wrote the paper with contributions from all authors. SC and PA performed the

sulfate measurements and the provided quality-controlled sulfate dataset used here. MA, PL, AW, RK contributed to the development of the GAW station CHC and to the overall planning for the SALTENA campaign. FB, VAS and RK conceived the study and led the overall scientific investigation.

*Competing interests.*  The authors declare no competing interests.

*Disclaimer.*  The authors declare no conflict of interest.

*Acknowledgements.*

– We acknowledge CSC – IT Center for Science Ltd. for the generous allocation of computational resources.

– DA and FB are funded by ERC (project CHAPAs no. 850614) and Finnish Centre of Excellence as well as the Academy of Finland (project no. 311932, 315203 and 337549).

– This study was supported by UMSA (Universidad Mayor de San Andrés) through the Laboratory for Atmospheric Physics (LFA) part
of the Institute for Physics Research. The LFA provided scientific, administrative and logistical support.

– We also acknowledge the financial, administrative and logistical support from IRD (Institut de Recherche pour le De´veloppement) representation in La Paz Bolivia, through funding from by Labex OSUG@2020.

– PA acknowledges funding from FAPESP - Fundação de Amparo à Pesquisa do Estado de São Paulo trough grant 2017/17047-0.

– DA wishes to thank the many scientists, technicians and personnel involved in the Chacaltaya station whose outstanding commitment
enables high-quality atmospheric records in a challenging environment.

– DA thanks Sara Blichner for valuable input during the writing process.

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
