# Peer review of "Identifying source regions of air masses sampled at the tropical high-altitude site of Chacaltaya using WRF-FLEXPART and cluster analysis"

_Atmospheric Chemistry and Physics, 2021_

## Referee Comment (RC2)

**Review of**
**Identifying source regions of air masses sampled at the tropical high-altitude site of Chacaltaya using WRF-FLEXPART and cluster analysis**
**by Diego Aliaga et al.**

**General comments**

The work presented in this manuscript may be divided into two major parts:

1. WRF simulations and subsequent FLEXPART-WRF calculations to determine the influence regions for the Chacaltaya station, with hourly resolution at the receptor.

2. Regridding, clustering and interpretation of the results.

I think that the first part is very valuable and mostly well done (for a few issues I noted, see below).

As for the second part, my impression is that it does not fully exploit the information produced in the first part. The complicated postprocessing rather hides features rather than making them more easy to grasp. I cannot see a real benefit from the clustering process, at least in the way it was done and presented. The presentation quality of the clustering is insufficient.

Furthermore, I am afraid that the influence from the La Paz urban agglomeration is not represented as well as the original data would allow. It is obviously an important topic for the Chacaltaya measurements and should be given more attention throughout.

My suggestion would be to either completely replace or to amend the evaluation based on clustering by a direct statistical evaluation of SRR fields, detailling when which regions influence the monitoring station. Rather than a posteriori looking at which kinds of biomes are represented in which cluster areas, one could divide the total domain into regions of interest with similar trace substance emission properties (both natural and anthropogenic!), and then evaluate with respect to these regions, circumventing the complicated clustering process. If the clustering is still kept, the presentation quality both of the method and the results should be improved.

More detailed comments can be found below.

**Detailed comments**

1. Literature review: While the introduction discusses a number of relevant papers, it could include some more, especially for work in the Alps, including that related to mountain peak stations such as Jungfraujoch and Sonnblick.

2. Line 102: It should be explained why six months are sufficient to capture the climate of the area, and how representative the chosen period is with respect to interannual variability (ENSO).

3. Line 150: It would be better to give more details on the comparison with observations rather than to just claim "reasonable agreement". At least, some quantitative scores need to be provided to back up this claim.

4. Line 158: Why the choice of 4 d back? How far do you get with that, or in other words, how many of the particle trajectories would end inside the evaluation area? How much of the variability of the atmospheric components of interest can be explained by that?

5. Line 169: Why 500 m for the lowest evaluation layer? The boundary layer height at nighttime could be considerably lower than the 500 m layer. Then, the sensitivity does not represent the influence of surface emissions properly. On the other hand, a resolution of 500 m is much too dense for the upper layers.

6. Line 190: I don't think that you should neglect the meridian convergence in your domain: the 3% error that you cite (it might be a little more in the extreme case) translates to 30 km for a deviation of 1000 km from your central meridian which is on the order of your grid resolution (38 km). It would not have been such a computational burden to use a more accurate formula. In addition, if the resolution of the two grids is on the same order of magnitude, the regridding is not a trivial process. Therefore, even though a reference for the method is given, its key features should be described.

7. Line 198: It would be useful to provide the actual value of $e^\alpha$.

8. Line 200: Is the resolution of the innermost rings sufficient to properly resolve the influence of the urban agglomeration? I think that it might not be.

9. Line 214: What is the rationale for the smoothing? Unless your FLEXPART output fields are very patchy (in which case one should increase the particle number and/or decrease the output grid resolution - see remark above about vertical resolution), it would not be beneficial. As you are doing this on the polar grid, at larger distances where patchy output is more likely, the grid resolution is already decreased. On the other hand, if you apply smoothing in the near field where strong gradients occur and are important to be represented properly (urban emissions!), you deliberately worsen your data.

10. Line 219: It is not clear to me how you applied the normalisation. What is in the denominator of the normalisation? Note that the total residence time of in each simulated release hour should be the same, except if particles leave the domain, but in that case, I don't think that normalisation would be proper. The actual residence time is what determines the concentration change in a grid cell, not a normlised one.

11. What is the effect of "filtering out" zero values, and how exactly does it work? Do you mean that you eliminate regions of your domain where you have only rarely nonzero SRR values over your whole period? That might be justified, at least unless there are strong emitters at such locations, which even if it is rarely the case should not be missed.

12. Line 255: First of all, I think that we should primarily be interested in the surface influence and not so much in the boundary-layer influence. It is at the surface where the emissions occur. At times when the layer considered is well mixed, it does not matter how thick you select it (you would normalise with the thickness so that you obtain emission sensitivity with respect to the area source). If it is not well mixed, you are making a mistake if you take too thick a layer, which would be a problem at nighttime, as you mention. The only case where considering the PBL seems to be warranted is PBL-related chemistry, but then a Lagrangian model is probably not a good tool anyway. Secondly, I don't think that we are interested in the relative SRR. As explained before, the total SRR should not be highly variable, so this normalisation will not have much influence. But what is more important, the contribution of emissions from some area of interest is independent of whether particles leave the domain (meaning lower SRRtotal) or not.

13. The choice of mixing ratio vs. concentration for source and receptor in FLEXPART is not discussed, even though at the high elevation of the receptor, and as sources (output grid) vary between sea level and >15 km, this is a potentially important topic.

14. Fig 5 and associated discussion: In line with what I explained above, I would think that absolute values of SRR for surface influence would be more interesting than normalised or detrended ones. I do agree that finding out for which fraction of the time (and when!) there was little surface influence, but again, this should be quantified using an absolute threshold, not a fractional one.

15. Line 313: This correlation expresses what I tried to explain in comment 12. If we had well-mixed conditions up to 1.5 km, we would have perfect correlation (actually, a 1:1 relationship if properly normalised). One has to be aware, however, that this correlation will be higher after long transport and much lower very close to the source.
I think that the regression formula does not serve any practical purposes and should be skipped.

16. Line 325: In addition to Fig. 4b I would like to see a plot of the average diurnal variation of the surface SRR. – Instead of "campaign", you might want to use "investigation period".

17. Figures 6 and 8: Caption illegible, thus I am not able to comment on it.

18. Figures 7, 9, 11: The time series should be presented as filled curves or bar plots to facilitate the interpretation, subfigures should be framed, and vertical lines as time markers be drawn. If my understanding is right that each hour on each day can belong to one cluster only, it would be better to stack all contributions on top of each other (stacked filled curves or stacked bars). This should more quickly show the seasonal variability. Also, it would probably be usefulto separate, also in this figure, in one way or another, the diurnal patterns from the longer-term (seasonal) evolution.

19. Methods and results of clustering: It is too difficult to understand the clustering method in detail and why it was chosen. One reason is that a formal description with symbols and formulae has been avoided and only verbal

explanations are included. I don't think that the concept of "pathways" is particularly helpful if there is no such thing as natural pathways, and rather a continous spectrum of air movements is found. The results presented don't appear to be very different from a simple division into sectors, thus one is wondering whether the application of this clustering method has real benefit for the understanding the situation. Also, we are never shown how a single cluster and its members looks like, we only see cluster centroids and homogeneously coloured regions.

20. Conclusions section: While the conclusions appear to make sense, it is difficult to identify them in the figures presented. This might be another hint that the way of evaluation and presentation should be improved.

---

## Author Comment (AC1)

**Review of**

**Identifying source regions of air masses sampled at the tropical high-altitude site of Chacaltaya using WRF-FLEXPART and cluster analysis by Diego Aliaga et al.**

The authors would like to thank the anonymous referees for reviewing this manuscript. We thank the positive feedback and appreciate the helpful and detailed comments and suggestions. Below, in blue, we repeat the referees' observations and our comments are in black.

**Anonymous Referee #1**

The current paper describes potential origins of air masses arriving at the Chacaltaya (CHC) atmospheric research station using high-resolution numerical weather predictions from the weather research and forcasting model (WRF), back-trajectories from the WRF version of the Lagrangian particle dispersion model FLEXPART, and the K-means clustering algorithm. The power of the method is illustrated using a straightforward example. It is a thorough and well-written manuscript which can be useful for analysis of data recorded at the location of CHC in the future. As such, I am happy to recommend publication when the comment below has been adressed.

Thank you for this positive comment.

The most difficult part in analysing results from any clustering algorithm often is the selection of the number of clusters. The work could be nuanced here as in this type of numerical algorithms, the robustness and performance should be the main driver in chosing this parameter. I believe this choice should be adressed more carefully.

The selection of 18 clusters by the authors is founded by two reasons. The first is that it represents a local maximum in the parameter scan of the silhouette average score. The second uses prior assumptions of the authors based on the interest in 2 vertical levels, 2 horizontal scales, and 4 wind directions which led to the expectation of a solution near  $2 \times 2 \times 4 = 16$  clusters. Although this reasoning is very intuitive, I wonder if it is not too reductive in nature.

For example, it implies that all atmospheric observatories should be able to identify about 16 clusters when performing a similar analysis, independent of their location.

Furthermore, with equally valid reasoning one can caluclate a preference to other numbers of clusters. For example, the identification of 6 pathways (equivalent to directions in the near-field), would suggest a solution at  $2 \times 2 \times 6 = 24$  clusters. This would prompt the further investigation of the 23 cluster solution which shows a local maximum in the silhouette average score. Alternatively, one can assume that the vertical levels are coupled to the horizontal scales as wind speed in the free troposphere is generally larger than those in the boundary layer. This coupling does not allow multiplication to figure out the number of combinations possible. The result would not be  $2 \times 2 = 4$  spatial ranges but e.g. 3: short range, medium range, and long range clusters (as found by the authors when analysing the 18 clusters). Using the 4 directions from the assumption, you would expect a solution at  $3 \times 4 = 12$  clusters, using the 6 major pathways instead of directions one would anticipate a solution at  $3 \times 6 = 18$  clusters.

The 18 cluster solution, as selected by the authors, is good because it adequately described the data and has a straightforward interpretation. If the authors think this is insufficient reasoning to select the number of clusters, they can apply alternative clustering algorithms to illustrate the robustness of their choice. In my opinion, no further reasoning should be brought forward when using only 1 clustering technique as it will inevitably be subject to speculation.

We fully agree with the reviewer here in that it is always very difficult to decide on the number of clusters. In our opinion the decision of the number of clusters should be based on three considerations: (1) a quantitative estimate such as the silhouette score, (2) applicability to the scientific question you are trying to answer and (3) practical aspect, for example, ease of use in future application of the resulting dataset. Based on (1), as the reviewer correctly identified, k=18 and k=23 are both viable options; however, given (3) above, we decided for practical reasons (e.g. easier to analyse and visualise) to select 18 over 23. Again, as the reviewer points out, this is a good and valid choice since it adequately describes the data and has a straightforward interpretation. We now add details of these different considerations to the revised manuscript in section 3.3 and attempt to clarify the reasoning behind our choice of the number of clusters.

Furthermore, in our study the silhouette scores are not very high which means we really have a more of a continuum rather than very clear clusters. This means the results are less sensitive to the choice of the number of clusters than in a situation where very clear clusters (e.g. a large and clear maximum in the silhouette score is present) exist. In our analysis the main location and characteristics of the clusters is not overly sensitive to small changes in the number of clusters (i.e 16 vs 18), however, changing the number of clusters does cause the boundaries between each cluster to change. We have now revised the 3rd bullet point

at the end of section 6 to incorporate some of these additional points related to the selection of the number of clusters.

**Anonymous Referee #2**

**General comments**

The work presented in this manuscript may be divided into two major parts:

- 1. WRF simulations and subsequent FLEXPART-WRF calculations to determine the influence regions for the Chacaltaya station, with hourly resolution at the receptor.
- 2. Regridding, clustering and interpretation of the results.

I think that the first part is very valuable and mostly well done (for a few issues I noted, see below).

Thank you for this positive comment.

As for the second part, my impression is that it does not fully exploit the information produced in the first part. The complicated postprocessing rather hides features rather than making them more easy to grasp. I cannot see a real benefit from the clustering process, at least in the way it was done and presented. The presentation quality of the clustering is insufficient.

Thank you for highlighting this. On reflection after reading these comments, we realised that we had not explained the technical details of the clustering clearly enough. We address this by revising section 3 and also by adding an Appendix to the manuscript where we add many more technical details. Secondly, we also realised we did not explain clearly in the manuscript why we want to cluster the data rather than using the high resolution output. The main reason is computational cost. Clustering and the data processing was used to reduce a huge dataset (almost 4 Tbyte) to a much much smaller size that is therefore much easier to use, analyze, interpret and distribute to other researchers and last but not least, it is good way how to follow and characterize air mass history evolution over a longer temporal span. In this case 6 months. We

believe that this is necessary for long term studies (as is presented here) but the raw high resolution output can be used for case studies. We now stress this motivation more clearly in the manuscript. We have added details to section 3.

Furthermore, I am afraid that the influence from the La Paz urban agglomeration is not represented as well as the original data would allow. It is obviously an important topic for the Chacaltaya measurements and should be given more attention throughout.

We address this in our response to comment #8 below. However, we would like to stress that the overall aim of this study was to identify where air masses observed at CHC originate from, beyond La Paz and not to determine the influence of La Paz on CHC. The influence of La Paz—when present—is very important, and can be identified with high precision using the SRR output from this study and combining it with in-situ observations (e.g. Fig. 5, Bianchi, F. et al. The SALTENA experiment: Comprehensive observations of aerosol sources, formation and processes in the South American Andes. Bulletin of the American Meteorological Society 1, 1–46 [2021]).

In our approach, we did not want to make any assumptions about where air masses originate from.

My suggestion would be to either completely replace or to amend the evaluation based on clustering by a direct statistical evaluation of SRR fields, detailing when which regions influence the monitoring station. Rather than a posteriori looking at which kinds of biomes are represented in which cluster areas, one could divide the total domain into regions of interest with similar trace substance emission properties (both natural and anthropogenic!), and then evaluate with respect to these regions, circumventing the complicated clustering process. If the clustering is still kept, the presentation quality both of the method and the results should be improved.

Thank you for these suggestions. We retain the clustering approach but now have improved the presentation of this. We have also added section 4.1 with a very general statistical analysis.

More detailed comments can be found below.

**Detailed comments**

• 1. Literature review: While the introduction discusses a number of relevant papers, it could include some more, especially for work in the Alps, including that related to mountain peak stations such as Jungfraujoch and Sonnblick.

- Thank you for pointing this out. Studies on Jungfraujoch were carefully considered in this work, for example Sturm et al. 2013, Brunner et al. 2012, Conen et. al 2012 and Herrman et. al 2015. However we did not mention Jungfraujoch by name in the text. In the revised manuscript we state more clearly which studies have been performed at Jungfraujoch and in other places (Lines 66 and 77).
- 2. Line 102: It should be explained why six months are sufficient to capture the climate of the area, and how representative the chosen period is with respect to interannual variability (ENSO).
  - Six months are not sufficient to capture the climate of the area to do so fully would require multiple years of observations and simulations. Unfortunately due to practical and financial constraints a multi-year observational campaign was not possible. The computational cost of these simulations was high and therefore the simulations were only performed to cover the 6 month campaign period, not a full year. Previously, we had noted this limitation of our study at the end of section 6 (Line 522): "Due to computational limitations, and given that the campaign only lasted for six months, we only simulated six months. For a more complete physical understanding, if computational resources allow, we recommend that a full annual cycle is always simulated even if the observational campaigns the model simulations support are shorter in duration". However, we now add additional text, including: "The analysis provides detailed air-mass information for the duration of the intensive period rather than a climatological description (this would require a multiple-year study)" (Line 137).
- 3. Line 150: It would be better to give more details on the comparison with observations rather than to just claim "reasonable agreement". At least, some quantitative scores need to be provided to back up this claim.
  - An in-depth verification is challenging due to the limited observations in this area. However, we do have observations from CHC which we have quantitatively compared to the WRF model output from the closest grid point (which in the model has a surface elevation of 5049 m, which is 111 m lower from the observation height). Using the hourly temperature data and the daily accumulated precipitation data we have computed the mean bias (MB), the mean absolute error (MAE) and the root mean square error (RMSE) following equations 5.1, 5.2 and 5.3 respectively in Chapter 3 of "Forecast Verification: A Practitioner's Guide in Atmospheric Science, Second Edition. Edited by Ian T. Jolliffe and David B. Stephenson. C 2012

John Wiley & Sons, Ltd. Published 2012 by John Wiley & Sons, Ltd." For temperature the MB=-0.42°C indicating that WRF predicts slightly cooler temperatures than observed, the MAE = 1.35°C and the RMSE= 1.73°C. When these values are compared to the standard deviation (s.d.) of the temperature from WRF (s.d. =  $2.11^{\circ}$ C) and the observations ( $2.61^{\circ}$ C), we can conclude that the average magnitude of the WRF forecast error is small relative to the variation. The corresponding values for daily accumulated precipitation are MB=-1.6 mm (WRF is slightly too dry on average), MAE= 3.5 mm and RMSE= 6.3 mm. However, as precipitation is very variable from day-to-day (s.d. = 5.0 mm in the observations compared to the mean observed value of 2.45 mm), we also compute the occurrence of hits, false alarms, misses, and correct negative values by a given precipitation threshold (i.e a contingency table) and the corresponding accuracy defined as the number of hits + the number of correct negatives divided by the total number of days. Hit is when both the model and observation have precipitation exceeding the threshold, correct negative is when the model and observation both have precipitation below the threshold. Miss is when there is precipitation exceeding the threshold in the observations but not in the model and false alarm is when the model has precipitation exceeding the threshold but the observations do not. The resulting values are:

| threshold | hit | Corr. neg | miss | False
alarm | accuracy |
|-----------|-----|-----------|------|----------------|----------|
| 0 mm      | 99  | 38        | 11   | 34             | 0.75     |
| 1 mm      | 42  | 77        | 35   | 28             | 0.65     |
| 5 mm      | 12  | 118       | 36   | 16             | 0.71     |

- We have now added text about these statistics to section 2 near the original Line 150; a reference to Bianchi, F. et al. The SALTENA experiment: Comprehensive observations of aerosol sources, formation and processes in the South American Andes. Bulletin of the American Meteorological Society 1, 1–46 (2021); and a detailed explanation to the supplementary material.
- 4. Line 158: Why the choice of 4 d back? How far do you get with that, or in other words, how many of the particle trajectories would end inside the evaluation

area? How much of the variability of the atmospheric components of interest can be explained by that?

• This is a good point and we thank the reviewer for highlighting it. If we assume an average wind speed of 10 ms-1, particles can travel 3456 km within 4 days. Within this radius, we will capture the influences from the city of La Paz / El Alto, the Amazon, volcanos, the altiplano and the Pacific Ocean. Figure 1 in this response shows that within 4 days, for the median hourly simulation (cases), the average particle spends 94% of its residence time (90.24 hours) within the domain. There are a few cases (less than 5%) where particles spend less than 60% of their residence time within the domain. These cases happen when strong winds are present in the domain. In this revised manuscript we add text stating that the average particle spends 94% of its residence time within the domain. In this revised manuscript we add text stating that the average particle spends 94% of its residence time within the domain. In this revised manuscript we add text stating that the average particle spends 94% of its residence time within the domain.

- Figure 1. Cumulative distribution function for each simulation hour (case) of the mean particle residence time.
- 0
- 5. Line 169: Why 500 m for the lowest evaluation layer? The boundary layer height at nighttime could be considerably lower than the 500 m layer. Then, the sensitivity does not represent the influence of surface emissions properly. On the other hand, a resolution of 500 m is much too dense for the upper layers.
  - We appreciate that the choice of 500m is somewhat subjective. Preliminary runs of FLEXAPRT were made with varying vertical levels but given the high variability of height of the terrain, this meant that particles traveling along the same pressure level—as it is often the case—were traversing different vertical grid resolutions as they moved horizontally making computations cumbersome and the output difficult to physically interpret.

- For this reason, we needed to use a vertical grid with a constant resolution throughout the whole depth of the atmosphere instead of the customary varying resolution a.g.l. so that comparison of vertical grids for locations with considerably different ground height a.s.l. is easier. For example consider the grid-cells above CHC (5 km a.s.l.) and La Paz (3.6 km a.s.l.). If we were to use varying vertical levels a.g.l. then an air mass moving along the same pressure level as CHC would move from a high resolution vertical grid to a low resolution vertical level in less than 20 km (the horizontal distance from CHC to La Paz). We selected a dz of 500~m as a compromise; ideally we want as small as dz as possible near the surface but to minimize computational cost we want a large as possible dz and thus fewer vertical levels. The constant dz also makes the conversion between a.s.l. and a.g.l. seamless.
- Given the lengthy explanation required regarding the choice of dz=500m, we have added this to an appendix along with additional technical details.
- 6. Line 190: I don't think that you should neglect the meridian convergence in your domain: the 3% error that you cite (it might be a little more in the extreme case) translates to 30 km for a deviation of 1000 km from your central meridian which is on the order of your grid resolution (38 km). It would not have been such a computational burden to use a more accurate formula. In addition, if the resolution of the two grids is on the same order of magnitude, the regridding is not a trivial process. Therefore, even though a reference for the method is given, its key features should be described.
  - This is a valid concern. Thank you for bringing it up. We want to highlight that the meridian convergence in this case produces mostly the same deviations found in the lat/lon grid which is commonly used near the tropics. That is, pixels near the equator cover more area than pixels near the poles. However there is no error in the location of these pixels. In our case it also implies that the rings in the log-polar ring are not perfectly equidistant from the center, they vary up to 3% which we consider to be small enough.
  - Regarding the regridding procedure from the lat-lon grid to the log-polar grid, we have now added the technical details of this to the appendix.
- 7. Line 198: It would be useful to provide the actual value of e
  - e is Euler's number (2.71). We have now added this to the text.
- 8. Line 200: Is the resolution of the innermost rings sufficient to properly re-solve the influence of the urban agglomeration? I think that it might not be.
  - The urban area of La Paz El Alto covers an area of 558km2. Since this is very close to the station of CHC and the center of our radial grid (but outside the area we have neglected, see figure A1 in the revised

manuscript), the grid cells are very small in this region - the typical area of the grid cells is ~15km^2. In total the urban area is covered by ~ 37 grid cells which we conclude is adequate to resolve the urban area. We now add text concerning this point to Line 628.

- To show the resolution of the log-polar grid, we have added a new figure (Fig A1) to the Appendix and also a detailed depiction of the location of La Paz (Fig. 4b).
- 9. Line 214: What is the rationale for the smoothing? Unless your FLEXPART output fields are very patchy (in which case one should increase the parti-cle number and/or decrease the output grid resolution see remark above about vertical resolution), it would not be beneficial. As you are doing this on the polar grid, at larger distances where patchy output is more likely, the grid resolution is already decreased. On the other hand, if you apply smoothing in the near field where strong gradients occur and are important to be represented properly (urban emissions!), you deliberately worsen your data.
  - Thank you for pointing out the unexplained parts of our method. We have realized that a proper explanation of the clustering methods is missing and we have now added it to the appendix.
  - Regarding the smoothing in particular, we would like to point out that the smoothing and normalization are applied just for the sake of clustering the cells. After the clusters are obtained (i.e. once each grid cell has been allocated a cluster number), the SRR values of these clusters are obtained from the original (not smoothed and not normalised) log-polar array of SRR.
  - Clustering without smoothing produces very similar results except that when smoothing is not used, very low and intermittent influence cells are assigned clusters randomly rather than matching their neighbours. The smoothing forces these few 'problematic' cells to be assigned to the neighboring group.
- 10. Line 219: It is not clear to me how you applied the normalisation. What is in the denominator of the normalisation? Note that the total residence time of in each simulated release hour should be the same, except if particles leave the domain, but in that case, I don't think that normalisation would be proper. The actual residence time is what determines the concentration change in a grid cell, not a normlised one.

- Thank you for pointing out that the explanation of the normalization procedure was somewhat ambiguous.
- We applied two different types of normalization: quantile normalization (Line 219) and total SRR normalization (Line 270). The purpose of using these two types of normalisation is very different. We feel that some confusion may have arisen given that our explanation was somewhat ambiguous which we now attempt to resolve.
  - The quantile normalization is solely used for clustering purposes as described in the answer to comment 9. When clustering, normalization of the dataset is a common and important procedure so that the distance metric on the clustering algorithm can correctly specify how similar (or dissimilar) different elements are. In our case, we use quantile normalization because it produces similar results to MinMax normalization but is also robust to extreme values and thus performs better when assigning groups to different cells. In the revised manuscript we explain the technical details and reason for using quantile mapping in the new appendix.
  - The second normalization (Line 256, SRR %) normalizes the SRR over the theoretical maximum total SRR of the simulated period. The theoretical maximum total SRR is the SRR that we would obtain if we added all the SRR's of each individual cell. In our case, since we simulate 96 hours back in time, the theoretical maximum is 3600 [seconds] x 96 [hours] = 345 600. This is a constant value. This was explained in Line 272, however, we now move this after equations 2 and 3 (Lines 280 and 285).
- 11. (Line 220) What is the effect of "filtering out" zero values, and how exactly does it work? Do you mean that you eliminate regions of your domain where you have only rarely nonzero SRR values over your whole period? That might be justified, at least unless there are strong emitters at such locations, which even if it is rarely the case should not be missed.
  - Thank you for pointing this out. Yes, this means that we eliminate grid cells where we often have zero SRR (c.f. where we rarely have non zero SRR values). This does not often happen in the surface grid cells but is much more common in the grid cells high above ground. The technical details of how this is done are explained in Lines 221 to 222.
  - However we now add a complementary explanation in the Appendix.
- 12. Line 255:
  - First of all, I think that we should primarily be interested in the surface influence and not so much in the boundary-layer influence. It is at the

surface where the emissions occur. At times when the layer considered is well mixed, it does not matter how thick you select it (you would normalise with the thickness so that you obtain emission sensitivity with respect to the area source). If it is not well mixed, you are making a mistake if you take too thick a layer, which would be a problem at nighttime, as you mention. The only case where considering the PBL seems to be warranted is PBL-related chemistry, but then a Lagrangian model is probably not a good tool anyway.

- In this study, and in other on-going work about the SALTENA campaign, we are equally interested in two main air masses: (1) air masses which have been influenced by the surface and (2) free tropospheric, ("clean") air masses. The reason we also calculate the pseudo BL influence is that the percentage of air not influenced by the lowest 1500m can be assumed to represent the free troposphere (FT). e.g. previously in Line 317 we stated: "Indirectly this means that approximately 76% of the air sampled at CHC can be considered representative of the FT". We are of the opinion that this is a more accurate estimate of the FT than the percentage of air not influenced by the surface (lowest 500m). To make this clearer, we have revised the text beneath equations 2 and 3 in the new manuscript.
- Secondly, I don't think that we are interested in the relative SRR. As explained before, the total SRR should not be highly variable, so this normal-isation will not have much influence. But what is more important, the contribution of emissions from some area of interest is independent of whether particles leave the domain (meaning lower SRRtotal) or not.
  - We agree that it would be arbitrary to have a total SRR based on the particles that remain in the domain. However, our normalization is performed using a constant theoretical maximum SRR (in other words SRR total = 4days\*24\*hours\*3600seconds) as stated in Line 273-276 (also see our response to comment #10 above). The conversion between percentage and absolute SRR is then a matter of multiplying/dividing by SRRtotal. Depending on the context then, it would make sense to use the absolute value or percentage. However, given our intended audience, we feel that Figure 4a (which in the revised manuscript is Fig. 5a) is better presented as a percentage as this is more understandable and meaningful than the absolute values of SRR. However we have also added the residence time to the figure so that the comparison between the two is better illustrated.

- 13. The choice of mixing ratio vs. concentration for source and receptor in FLEXPART is not discussed, even though at the high elevation of the receptor, and as sources (output grid) vary between sea level and >15 km, this is a potentially important topic.
  - Thank you for highlighting this. In Line 160 we mentioned that the units of SRR are in seconds (output units). However now we are also including (in the same Line) the use of options ind\_receptor=2 and ind\_source=2 (meaning that we are using mixing ratios both for the receptor and the source).
- 14. Fig 5 and associated discussion: In line with what I explained above, I would think that absolute values of SRR for surface influence would be more interesting than normalised or detrended ones. I do agree that finding out for which fraction of the time (and when!) there was little surface influence, but again, this should be quantified using an absolute threshold, not a fractional one.
  - We think the reviewer is referring to the original Fig. 4b (not 5) in the manuscript as otherwise the topics are different.
  - We have now revised the original Figure 4 (now Figure 5) based on the reviewers comments. We do keep the normalised SRR and we hope that now having explained the normalisation more clearly (see point #10 above) this is justified. To make the values more understandable, we have added a second x-axis to Figure 5a, and a second y-axis to Figure 5c, showing the hours. Notice that 100% = 96 hours so the conversion between percentage and hours in practice is 1% ~= 1 hour (1.00 % = 1.04 hours).
  - However, we have replaced the detrended surface SRR with the non detrended values as we agree this is easier to interpret.
- 15. Line 313: This correlation expresses what I tried to explain in comment 12. If we had well-mixed conditions up to 1.5 km, we would have perfect correlation (actually, a 1:1 relationship if properly normalised). One has to be aware, however, that this correlation will be higher after long transport and much lower very close to the source. I think that the regression formula does not serve any practical purposes and should be skipped.
  - We are also interested in detecting periods of mostly clean free tropospheric air masses. We create the pseudo boundary layer so that we can better define the free troposphere (see also our response to comment #12 above). We have also now removed the regression equation and just report the slope value and correlation coefficient in the text.

- 16. Line 325: In addition to Fig. 4b I would like to see a plot of the average diurnal variation of the surface SRR. – Instead of "campaign", you might want to use "investigation period".
  - Thank you for the suggestion. We have added a new panel (Fig 5c) with the average SRR diurnal variation.
- 17. Figures 6 and 8: Caption illegible, thus I am not able to comment on it.
  - We are sorry that these captions appear illegible. On the pdf available from the ACPD website these appear visible but we will double check our files. Furthermore, we have copied the text of these two captions here to help the reviewer.
    - Figure 8. Heat map showing the biome and land cover characteristics associated with each of the 18 clusters. The percentage values indicate the percentage of air in each cluster that travels over each different biome. The darker the colour, the stronger the influence from the corresponding biome. For a description of the biomes see Section 4. Due to rounding errors and the use of integers, some columns add to 99 rather than 100.
    - Figure 6. Centroid properties for each of the main pathways (PW).
      Panel a) shows the median height above ground level of each cluster while panel b) shows the median height above sea level.
      Panel c) shows the ratio between the SRR values that are below .5 km above ground level and the total SRR value for each cluster.
      Panel d) shows the mean SRR percentage for each PW.
      Quantitative numbers are presented in Fig. S1.

 18. Figures 7, 9, 11: The time series should be presented as filled curves or bar plots to facilitate the interpretation, subfigures should be framed, and vertical lines as time markers be drawn. If my understanding is right that each hour on each day can belong to one cluster only, it would be better to stack all contributions on top of each other (stacked filled curves or stacked bars). This should more quickly show the seasonal variability. Also, it would probably be useful to separate, also in this figure, in one way or another, the diurnal patterns

**from the longer-term (seasonal) evolution.**

- We have now re-made Figures 6, 7, 9, 10, and 11 following the reviewers suggestions and note that the exact values of the data displayed in these figures is also reported in Figure S1 and Figure 12.
- There is a misunderstanding here: "each hour on each day can belong to one cluster only," - this is not correct. We do not cluster hours (times); we cluster grid cells and therefore each hour can have contributions for more than one cluster. Physically, this means that air sampled at CHC at one hour can originate from more than one location - a process that is possible due to the chaotic and turbulent nature of the atmosphere. This is mentioned in the abstract, Line 11: "A key aspect of our method is that it is probabilistic and for each observation time, more than one air mass (cluster) can influence the station and the percentage influence of each air mass can be quantified". However to ensure that the clustering is correctly understood we have revised section 3 and added more technical details to an appendix.
- Regarding the diurnal patterns, we previously decided to show them in the supplementary material: Figure S3 is a power spectra of all clusters which identifies clusters 02\_SR, 07\_SR, 10\_SR and 12\_SR as potential candidates for presenting a diurnal pattern. These diurnal patterns are then shown in Figures S4 to S7. We have not found a way to include these results in Figures 7, 10 and 11 without cluttering the figures so we keep them in the supplementary material.
- 19. Methods and results of clustering: It is too difficult to understand the clustering method in detail and why it was chosen. One reason is that a formal description with symbols and formulae has been avoided and only verbal explanations are included. I don't think that the concept of "pathways" is particularly helpful if there is no such thing as natural pathways, and rather a continuous spectrum of air movements is found. The results presented don't appear to be very different from a simple division into sectors, thus one is wondering whether the application of this clustering method has real benefit for the understanding the situation. Also, we are never shown how a single cluster and its members looks like, we only see cluster centroids and homogeneously coloured regions.
  - Thank you for sharing this insight with us, we attempted to keep the manuscript short but now thanks to the reviewer's comments we realise that additional details, and mathematically details are needed and have added them to an appendix.
  - For practical reasons, we wanted to produce a data set with a small number of clusters. We selected 6 based on the silhouette score. We struggled to come up with a descriptive name for the 6 clusters which

would be clearly different from the 18 clusters. We chose "climatological pathways" as it is a term used by Fleming et al in a review of source area identification methods.

- These 6 pathways (and by extension the 18 clusters) are in a sense a simple division into 3D sectors but with the boundaries—both vertically and horizontally—between them identified objectively rather than purely subjectively.
- Related to the point about cluster centroids, we have added two new figures (Figs. S9 and S10) to the supplementary material which show the location and SRR of all pathways and clusters at different height levels. These figures also show the vertical and hence 3-dimensional aspects of the different clusters. For example Figure S8 shows that not all pathways are present at all vertical levels and hence the clustering identifies more complex cluster areas than a simple manual sector analysis. We refer to these new figures in section 5.3 and 5.4.
- 20. Conclusions section: While the conclusions appear to make sense, it is difficult to identify them in the figures presented. This might be another hint that the way of evaluation and presentation should be improved.
  - Hopefully with the revisions made to the paper, now it is easier to find the evidence to support our conclusions. However, to help a reader on this point, for each bullet point in the conclusions section of the revised manuscript we now refer to the figure where the evidence can be found.

| ∎ FIG | 5a |
|-------|----|
|-------|----|

- FIG 5b
- FIGS 6,8,11,12,S2
- FIG 7
- FIG 7,10
- FIG 10
- FIG 10
- FIGS 8,11,12,S4,S5,S6,S7.